# Characterization of a selective, iron-chelating antifungal compound that disrupts fungal metabolism and synergizes with fluconazole

Jeanne Corrales,[1,2,3] Lucia Ramos-Alonso,[3] Javier González-Sabín,[4] Nicolás Ríos-Lombardía,[4] Nuria Trevijano-Contador,[5] Henriette Engen Berg,[6] Frøydis Sved Skottvoll,[6] Francisco Moris,[4] Oscar Zaragoza,[4,7] Pierre Chymkowitch,[3] Ignacio Garcia,[8] Jorrit M. Enserink[1,2,3]

**ABSTRACT**    Fungal infections are a growing global health concern due to the limited number of available antifungal therapies as well as the emergence of fungi that are resistant to first-line antimicrobials, particularly azoles and echinocandins. Development of novel, selective antifungal therapies is challenging due to similarities between fungal and mammalian cells. An attractive source of potential antifungal treatments is provided by ecological niches co-inhabited by bacteria, fungi, and multicellular organisms, where complex relationships between multiple organisms have resulted in evolution of a wide variety of selective antimicrobials. Here, we characterized several analogs of one such natural compound, collismycin A. We show that NR-6226C has antifungal activity against several pathogenic *Candida* species, including *C. albicans* and *C. glabrata*, whereas it only has little toxicity against mammalian cells. Mechanistically, NR-6226C selectively chelates iron, which is a limiting factor for pathogenic fungi during infection. As a result, NR-6226C treatment causes severe mitochondrial dysfunction, leading to formation of reactive oxygen species, metabolic reprogramming, and a severe reduction in ATP levels. Using an *in vivo* model for fungal infections, we show that NR-6226C significantly increases survival of *Candida*-infected *Galleria mellonella* larvae. Finally, our data indicate that NR-6226C synergizes strongly with fluconazole in inhibition of *C. albicans*. Taken together, NR-6226C is a promising antifungal compound that acts by chelating iron and disrupting mitochondrial functions.

**IMPORTANCE**    Drug-resistant fungal infections are an emerging global threat, and pan-resistance to current antifungal therapies is an increasing problem. Clearly, there is a need for new antifungal drugs. In this study, we characterized a novel antifungal agent, the collismycin analog NR-6226C. NR-6226C has a favorable toxicity profile for human cells, which is essential for further clinical development. We unraveled the mechanism of action of NR-6226C and found that it disrupts iron homeostasis and thereby depletes fungal cells of energy. Importantly, NR-6226C strongly potentiates the antifungal activity of fluconazole, thereby providing inroads for combination therapy that may reduce or prevent azole resistance. Thus, NR-6226C is a promising compound for further development into antifungal treatment.

**KEYWORDS**    antifungal, *Candida*, iron chelator, metabolism, transcriptomics, antibiotic resistance

Address correspondence to Ignacio Garcia, ignacio.garciallorente@fhi.no, or Jorrit M. Enserink, jorrit.enserink@ibv.uio.no.

F.M. is employed by the biotechnology company EntreChem SL, which provided several compounds used in this study.

See the funding table on p. 17.

*[This article was published on 17 January 2024 with an incorrect affiliation for Pierre Chymkowitch. The affiliation was corrected in the current version, posted on 22 January 2024.]*

Fungal infections range from relatively benign infections of the skin and mucosal tissues to life-threatening invasive infections, affecting more than a billion people worldwide and killing over 1.5 million people annually (1). Infections with *Candida*

species, such as *C. albicans* and *C. glabrata*, are among the most common human fungal infections (1). Among the main antifungals that are used in the clinic today are azoles, echinocandins, and polyenes (2). Azoles block cell membrane synthesis by inhibiting the ergosterol biosynthesis enzyme, lanosterol 14-α-demethylase (encoded by *ERG11* in *Candida*), whereas echinocandins disrupt cell wall synthesis via noncompetitive inhibition of the (1,3)-β-d-glucan synthase enzyme encoded by *FKS* genes. However, long-term use of antifungals has resulted in a steadily increasing prevalence of antifungal resistance, which is a major global health concern (3). Therefore, there exists a need for new antifungal drugs for treatment of antifungal-resistant *Candida* and other pathogenic fungal species.

During evolution, dynamic interactions that occur in ecological niches co-inhabited by bacteria and fungi have resulted in formation of a wide range of antimicrobial compounds that are a rich source of potential antifungal treatments (4). One example of a complex multi-organism interaction occurs in colonies of leafcutter ants, which form a mutually beneficial symbiotic relationship with a specialized fungus and with various antimicrobial-producing bacteria, including *Streptomyces* sp. The ants provide freshly cut leaves to the fungus, resulting in a fungal garden. The fungi produce specialized fungal structures that serve as a key food source for the ant larvae (5). The bacteria, which grow on the cuticle of the ants, synthesize antifungal compounds such as candicidin or nystatin that protect the fungal garden from invasion by parasitic fungi, which can ruin the fungal garden and destroy the colony (6, 7). Another such compound is collismycin A (ColA; Fig. 1A), which is produced by *Streptomyces* and which has antimicrobial activity against several fungi, including *C. albicans* (8). However, ColA also possesses substantial cytotoxic activities against mammalian cells (9), rendering it less useful as an antifungal therapy.

In this study, we characterized the antifungal activity and overall toxicity of several ColA analogs and identified a compound with antifungal activity against clinical isolates of azole- and echinocandin-resistant *Candida* species but with reduced toxicity against mammalian cells. Our results suggest that this ColA analog is a promising candidate for development into a treatment for infections caused by wild-type (WT) and drug-resistant *Candida* species.

## RESULTS

### Identification of collismycin A analogs with increased antifungal activity

To investigate the potential use of novel ColA analogs as selective antifungal agents, we generated a compendium of 26 collismycin-related compounds (Fig. S1A; see Materials and Methods for details). To determine selectivity against pathogenic fungi, we screened these compounds against a panel of *Candida* strains and two human cell lines. We used *Candida albicans* (CCUG32723), *Candida glabrata* (ATCC15545), as well as an echinocandin-resistant *C. glabrata* strain (*fksI-L662W*; JEY12725) and a fluconazole-resistant *C. glabrata* strain (FL-256; JEY12726), both of which were isolated at Oslo University Hospital (Table S1). First, we studied the proliferation of these fungi on increasing concentrations of the starting compound, ColA. ColA reduced the proliferation of all pathogenic yeasts tested (Fig. 1B; Fig. S2 and S3A). We then tested the other analogs and found that the antifungal activity was lost for most of the compounds (Fig. S2). However, three compounds, NR-6226C, NR-6226K, and NR-6226V, inhibited the growth of *Candida* spp. more potently than ColA (Fig. 1C; Fig. S3A and B). NR-6226C appeared to be the most efficacious, yielding a half maximal effective concentration ($EC_{50}$) at least 12 times smaller than ColA-treated strains, whereas treatment with either NR-6226K or NR-6226V resulted in an inhibitory effect that was approximately six times greater than ColA (Fig. S3B). Importantly, the two clinically isolated antimicrobial-resistant *C. glabrata* strains were also sensitive to these compounds (Fig. 1C; Fig. S3B). These results encouraged us to further explore the potential of these drugs for further development into antifungal therapy.

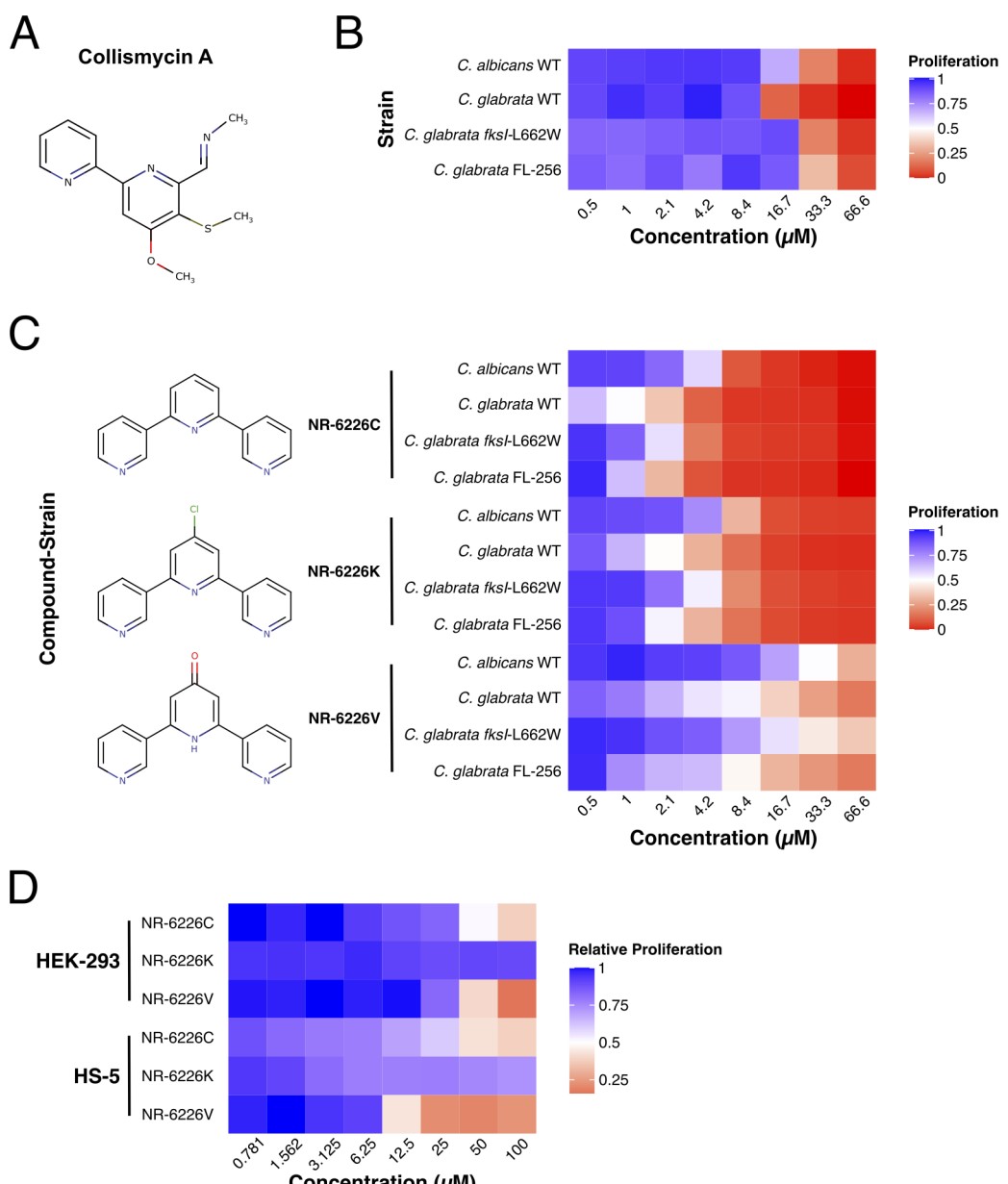

**FIG 1** Identification of the ColA analog NR-6226C as an antifungal agent with a potential therapeutic window. (A) ColA compound structure. (B) Heat map of relative proliferation of several wild-type (WT) and echinocandin- and azole-resistant *Candida* strains. The indicated strains were incubated in synthetic defined medium (SD) with increasing concentrations of ColA for 24 h, after which proliferation was analyzed by measuring the optical density ($OD_{600}$). Data were normalized to dimethyl sulfoxide (DMSO)-treated control samples. (C) Heat map of *Candida* proliferation in the presence of the ColA analogs NR-6226C, NR-6226K, and NR-6226V. The indicated fungal strains were incubated in the presence of increasing compound concentrations in yeast nitrogen base (YNB) −iron (0.69% yeast nitrogen base without amino acids or iron, with 2% dextrose) media for 24 h, after which proliferation was measured using $OD_{600}$. (D) Human HEK-293 and HS-5 cells are considerably less sensitive to ColA analogs than *Candida* spp. Cells were incubated with increasing concentrations of the indicated compounds for 24 h, after which relative cell viability was assayed using CellTiter-Glo. Data were normalized to DMSO controls.

For any novel antifungal compound to have a clinical application, it is essential that there exists a therapeutic index, i.e., fungi should be more sensitive to the compound than mammalian cells. Therefore, we tested the sensitivity of HS-5 and HEK-293 fibroblast cell lines to NR-6226C, NR-6226V, and NR-6226K. $EC_{50}$ values of NR-6226C were 37.05 ± 6.94 µM and 28.68 ± 8.32 µM in HEK-293 and HS-5 cell lines, respectively (Fig. 1D; Fig.

S3C), and although we noticed morphological changes at very high concentrations of NR-6226C (100 µM; Fig. S4), these data suggest the existence of a potential therapeutic window. Treatment with NR-6226K appeared to have no discernible effects on the proliferation of either cell lines; however, closer inspection revealed precipitation of the compound (Fig. S4), rendering NR-6226K less useful in physiological settings. We found that NR-6226V strongly attenuated cell proliferation of HS-5 cells ($10.49 \pm 1.42$ µM) and, to a somewhat lesser extent, also that of HEK-293 cells, limiting its usefulness as a potential antifungal treatment (Fig. 1D; Fig. S3C). For the remainder of the study, we decided to focus on NR-6226C due to its superior activity compared to ColA, its better solubility, its favorable toxicity profile for human cells, and its effectiveness against antifungal-resistant *C. glabrata* strains.

## ColA analog NR-6226C is a $Fe^{2+}$ chelator

ColA has been previously shown to have iron-chelating properties through the formation of a complex containing two ColA molecules bound to a single atom of either $Fe^{2+}$ or $Fe^{3+}$, and the three nitrogen atoms in ColA have been proposed to facilitate binding of the iron atom to form a tridentate chelator (9–11).

To test whether iron chelation is important for the inhibitory effect of ColA on fungal growth, we tested the antifungal activity of collismycin H (ColH), which is a ColA derivative that contains a hydroxymethyl instead of a methyl-imine moiety and, therefore, cannot chelate $Fe^{2+/3+}$ (Fig. 2A, right). As shown in Fig. 2A, ColH lacked antifungal activity even at very high concentrations, suggesting that the three nitrogen atoms in ColA are essential for antifungal activity. While not conclusive by themselves, these data do support the idea that ColA antifungal activity and its derivatives are mediated through iron chelation. Given that some metal chelators can be promiscuous and bind different metals with different affinities (12), we tested which metals can be bound by NR-6226C using high-performance liquid chromatography and mass spectrometry. These nano liquid chromatography experiments showed that similar to ColA, NR-6226C forms a 2:1 compound-iron complex, where it preferentially binds $Fe^{2+}$ in a concentration-dependent manner (Fig. 2B; Fig. S5). NR-6226C can also bind $Fe^{3+}$ but to a much lesser extent than $Fe^{2+}$ (Fig. 2B).

Next, we determined whether chelation of metal ions is important for antifungal activity of NR-6226C. We performed growth assays of *C. glabrata* with NR-6226C in the absence or presence of 5 µM of various metal salts. As expected, the addition of $FeCl_2$ significantly increased the $EC_{50}$ of NR-6226C (Fig. 2C). The same was observed for $FeCl_3$, which is likely due to the fact that $Fe^{3+}$ can be converted into $Fe^{2+}$ by ferric reductases. Surprisingly, $Cu^{2+}$ and $Zn^{2+}$, which hardly bind NR-6226C even at high concentrations, also significantly increased the $EC_{50}$ of NR-6226C. The ameliorating effect of $Cu^{2+}$ and $Zn^{2+}$ on inhibition of fungal growth by NR-6226C may be caused by mismetallation, i.e., inactivation of a protein due to binding of a non-cognate metal (13, 14), which has previously been shown to induce a compensatory response that promotes uptake of $Fe^{2+}$ [15]. Overall, these findings are consistent with the idea that NR-6226C inhibits the proliferation of *Candida* strains mainly through iron chelation, although we cannot exclude the possibility that it acts through alternative mechanisms *in vivo*.

## Treatment with NR-6226C induces an iron starvation response

We wished to gain more insight into the physiological response of fungal pathogens to NR-6226C treatment by studying changes in gene expression programs. To avoid potential secondary effects of long-term drug treatment, we treated *C. glabrata* cells for 1 h with 10 µM NR-6226C and analyzed changes in mRNA levels by RNA sequencing. For comparison, we also included treatment with 5 µM Dp44mT, which is a known iron chelator (15). We found that 224 genes were significantly upregulated and 220 genes were significantly downregulated upon NR-6226C treatment (Table S2). Interestingly, genes that were significantly upregulated by NR-6226C included genes involved in the response to iron starvation and the oxidative damage response, such as *TRR1*, *HMX1*, and

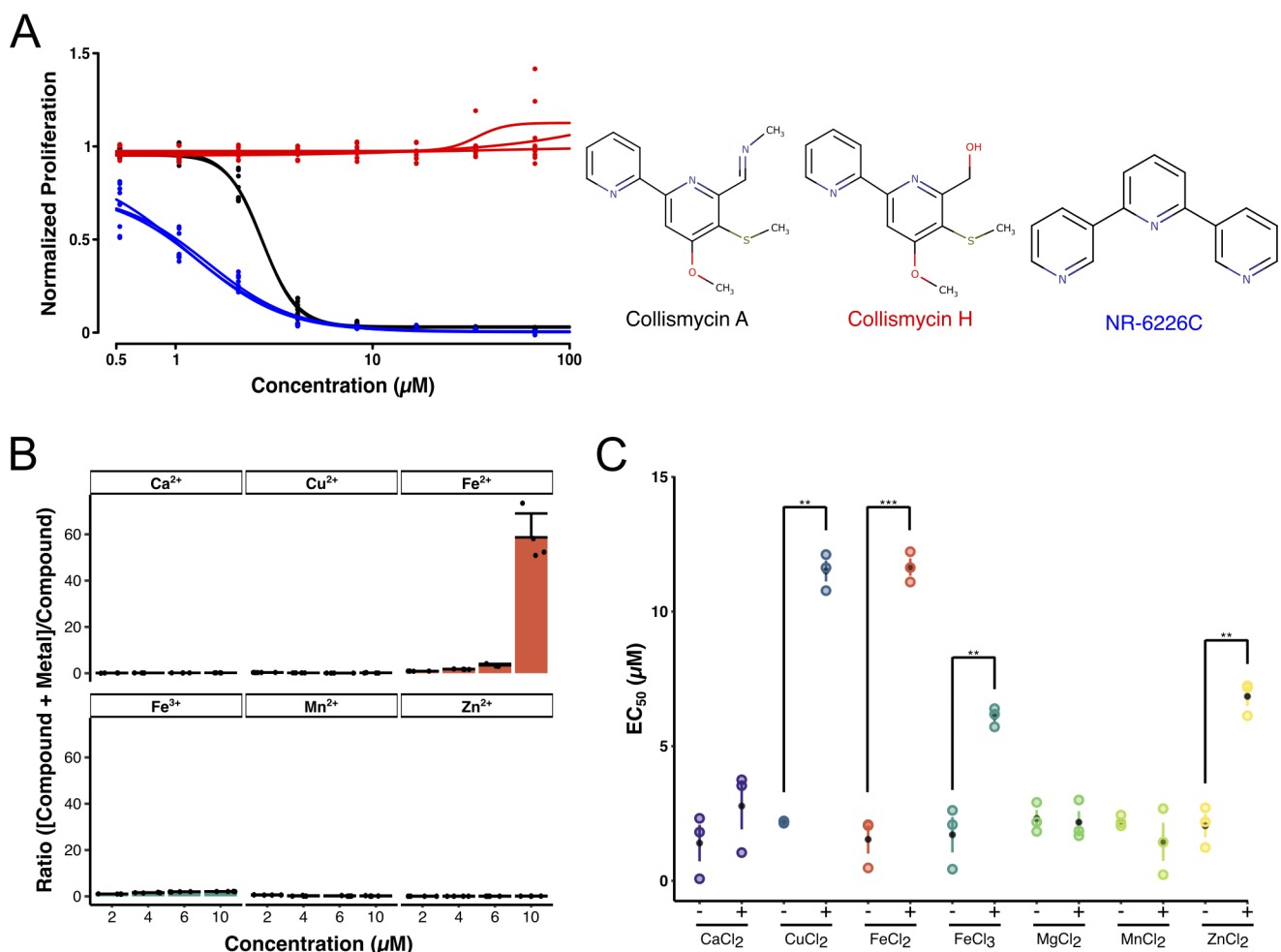

**FIG 2** NR-6226C is an iron chelator that inhibits *Candida* proliferation by sequestering iron. (A) ColH, which does not bind iron, does not inhibit growth of wild-type *C. glabrata* (JEY10028). Fungal cells were treated with ColA, ColH, and NR-6226C in SD media for 24 h, after which proliferation was measured using $OD_{600}$. Data were normalized to DMSO control samples. (B) NR-6226C selectively binds $Fe^{2+}$ *in vitro*. The interaction of NR-6226C with increasing concentrations of the indicated metal ions; $n = 3$. (C) Exogenous $Fe^{2+}$ rescues the antiproliferative effect of NR-6226C. Wild-type *C. glabrata* was incubated for 24 h with NR-6226C in presence of 5 µM $CaCl_2$, $CuCl_2$, $FeCl_2$, $FeCl_3$, $MgCl_2$, $MnCl_2$, or $ZnCl_2$, respectively. Proliferation was measured using $OD_{600}$, and values were normalized to DMSO controls. Resulting $EC_{50}$ values were calculated, and statistical significance was analyzed using unpaired two-sample *t*-tests in R; ***$P \leq 0.001$; **$P \leq 0.01$; $n = 3$.

*HBN1*, which encode thioredoxin, heme oxygenase, and an oxidoreductase, respectively (Fig. 3A and B). Other examples of genes induced by NR-6226C were *CAGL0G06798g*, which is a homolog of *Saccharomyces cerevisiae (Sc) LSO1* and known to be induced by iron starvation (16); *CAGL0L11990g*, a homolog of *Sc GRX4*, encoding an iron-response regulating enzyme with glutathione-dependent oxidoreductase and glutathione S-transferase activities (17); and *CAGL0D04708g*, a major copper transporter that is transcriptionally induced at low copper levels (18). Genes that were downregulated by NR-6226C include genes encoding proteins with iron-sulfur clusters, such as *CAGL0M00374g*, which encodes H2S-NADP oxidoreductase; the succinate dehydrogenase-encoding gene *SDH2* and the aconitase-encoding gene *ACO1*, which are preferentially expressed in the presence of sufficiently high iron levels (19). Other downregulated genes are *DUG3*, which mediates degradation of glutathione (20), and the detoxifying metallothionein gene *MT-I*, which is induced by high levels of metal ions (21). As expected, gene ontology (GO) analysis revealed that these downregulated genes function in mitochondrial and enzymatic functionalities, such as lyase activity, oxidoreductase activity, and catalytic activity (Fig. 3C). Together, these findings strongly suggest

that NR-6226C treatment induces an iron starvation response and possibly also an oxidative damage response.

To further characterize the transcriptional response to NR-6226C, we compared it to that of the known metal chelator Dp44mT, which chelates both copper and iron (15, 22). As expected, there was a substantial correlation between the transcriptional responses elicited by these two compounds, showing an overlap of 58% and 55% of genes that were significantly upregulated and downregulated, respectively (Fig. 3D through F). Genes encoding iron-binding metalloenzymes or enzymes containing iron-sulfur clusters were strongly downregulated by NR-6226C and Dp44mT treatment, such as *SDH2*, *ACO1/2*, *CAGL0E05676g* (*Sc TYW1*), *LYS9*, as well as the iron-sulfur assembly coding gene *CAGL0G03905g* (*Sc ISA1*). These results further support the idea that NR-6226C is a potent iron chelator.

The iron starvation response in *S. cerevisiae* has been reported to involve the transcription factors Aft1 and Aft2 (23–25). Given that *C. glabrata* is very closely related to *S. cerevisiae* (26) and because it is challenging to create gene deletions in *C. glabrata*, we used *S. cerevisiae* to test whether some of the responses to NR-6226C are mediated by Aft1. For comparison, we included Dp44mT as a control. Interestingly, treatment of WT *Sc* cells with NR-6226C resulted in robust activation of Aft1/2 target genes *FTR1*, *FET3*, *HMX1*, and *LSO1*, while there was no effect of NR-6226C on expression of control 5S RNA (Fig. 3G). Importantly, deletion of the *AFT1* gene attenuated the transcriptional responses upon treatment with NR-6226C (Fig. 3G).

Taken together, we conclude that treatment with NR-6226C induces a strong iron starvation response in *C. glabrata* and *S. cerevisiae*. Given that iron is a limiting factor in microbial pathogenesis (27), NR-6226C is a promising candidate for treatment of microbial infections.

## Treatment with NR-6226C induces reactive oxygen species (ROS) formation

Low levels of iron can result in dysregulation of iron-dependent processes, including oxidative phosphorylation, thereby leading to mitochondrial dysfunction and generation of ROS. Given our finding that NR-6226C treatment results in activation of genes involved in oxidative damage response, we tested whether NR-6226C induces formation of ROS in *C. glabrata*. Fungal cells were preloaded with the fluorescent ROS sensor $H_2DCF$-DA, after which we measured changes in fluorescence after 3, 6, and 24 h of exposure to dimethyl sulfoxide (DMSO), 30 µM NR-6226C, or 75 µM $H_2O_2$ (concentrations that strongly inhibited proliferation; Fig. S6A). Surprisingly, even though treatment with NR-6226C for 1 h induced the activation of genes involved in the oxidative damage response (Fig. 3A), no ROS formation could be detected after treatment for 3 or 6 h (Fig. 4A). Only after 24 h of NR-6226C treatment was significant ROS production observed. It is possible that short-term treatment with NR-6226C induces formation of low levels of ROS that are sufficient to induce transcriptional changes but that are below the detection limit of the $H_2DCF$-DA assay. It also cannot be excluded that NR-6226C activates the transcriptional oxidative damage response independently of ROS. Nonetheless, these data show that long-term exposure of fungal cells to NR-6226C can induce ROS production.

We hypothesized that NR-6226C-induced ROS formation was responsible for the antifungal effect of NR-6226C. However, treatment with the ROS scavenger N-acetyl-cysteine did not rescue the negative effect of NR-6226C on fungal cell viability (Fig. S6B), suggesting that ROS production is not the primary cause of inhibition of cell proliferation. Rather, inactivation of crucial iron-dependent metabolic processes may underlie the effect of NR-6226C, such as loss of ATP production and reduced synthesis of various essential metabolites. To test whether NR-6226C affects metabolism and cellular ATP production, we treated *C. glabrata* with increasing concentrations of NR-6226C during a 24-h time course. Equal numbers of cells were harvested for each concentration and time point, and relative ATP levels were quantified using CellTiter-Glo (28). As shown in Fig. 4B, treatment with NR-6226C resulted in a concentration- and time-dependent reduction in luminescence, reflecting decreased metabolic activity and loss of cellular ATP

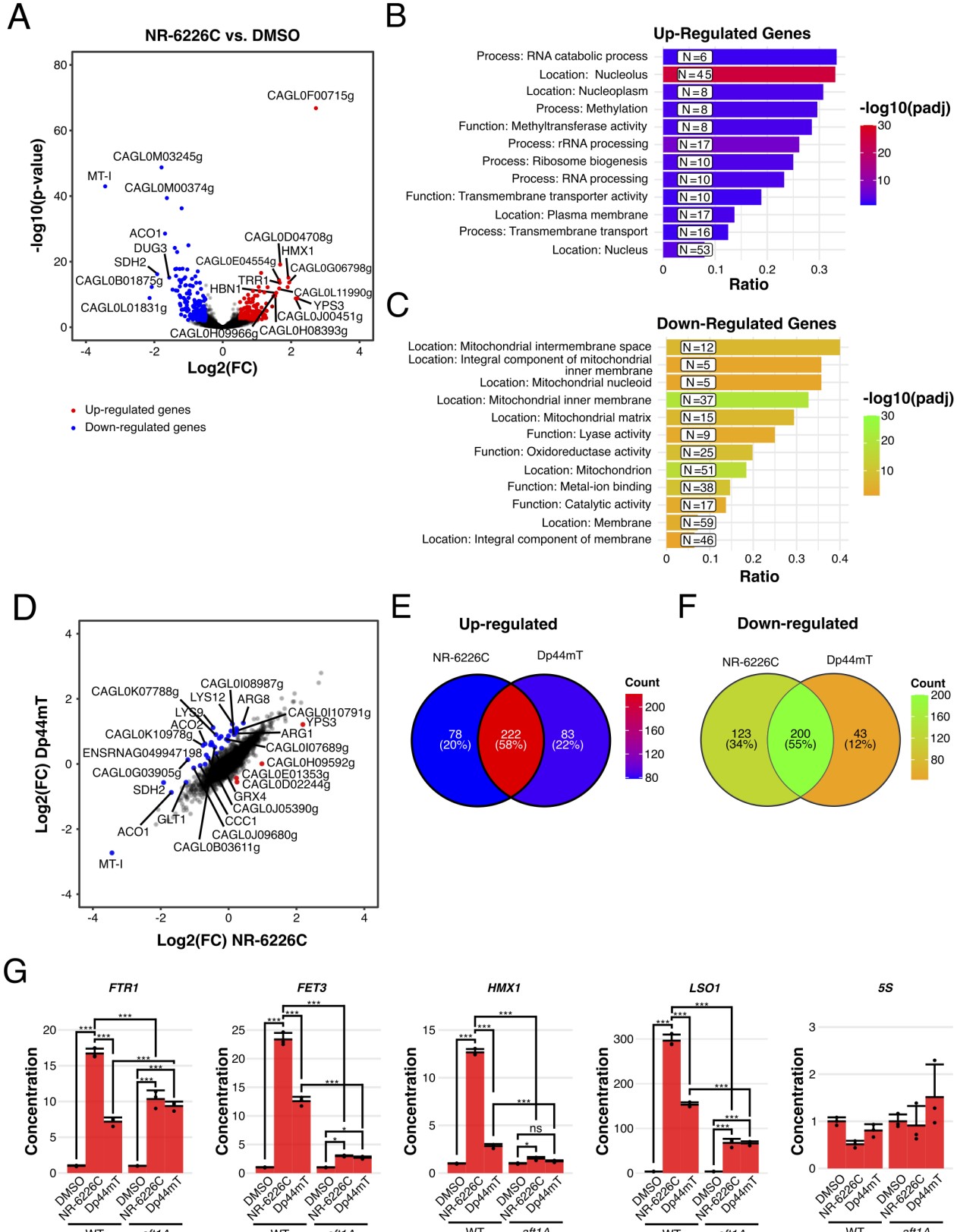

**FIG 3** NR-6226C treatment induces an iron starvation response. (A) Wild-type *C. glabrata* cells were incubated with either DMSO or 10 µM NR-6226C for 1 h, after which RNA levels were analyzed by RNA sequencing. The volcano plot shows transcripts below (blue) or above (red) a 1.4-fold change (log2 fold change <−0.5 or >0.5) and an adjusted *P*-value <0.01 threshold in NR-6226C-treated cells relative to the DMSO control. (B and C) GO analysis of genes that (Continued on next page)

**FIG 3** (Continued)

are either upregulated (B) or downregulated (C) by NR-6226C treatment. (D) Volcano plot showing commonly regulated genes based on the threshold for the calculated mean difference and standard deviation in log fold change between *C. glabrata* cells treated with either NR-6226C or Dp44mT (log2 fold change <−0.683 or >0.635). (E and F) Venn diagram showing the count and respective percentages of upregulated (E) or downregulated (F) genes between *C. glabrata* cells treated with either NR-6226C or Dp44mT. (G) Validation of a panel of selected genes by quantitative reverse transcription PCR (RT-qPCR) shows that NR-6226C treatment activates the transcription factor Aft1. Wild-type or *aft1Δ S. cerevisiae* strains were treated with DMSO, 10 μM NR-6226C, or 10 μM Dp44mT for 24 h, after which RNA levels were analyzed by RT-qPCR. cDNA concentrations were normalized to DMSO controls, followed by calculation of the means, sample standard deviations, and statistical significance using one-way analysis of variance (ANOVA) and Tukey's HSD for post hoc analyses in R. ***$P \leq 0.001$; **$P \leq 0.01$; ns, not significant; $n = 3$.

production. This suggests that treatment with NR-6226C has a negative effect on mitochondrial activity. To more directly measure mitochondrial activity, we used a colorimetric assay in which the yellow dye 2,3-bis(2-methoxy-4-nitro-5-sulfophenyl)-2*H*-tetrazolium-5-carboxanilide (XTT) is reduced by NADH by mitochondrial dehydrogenases (29). Consistent with the CellTiter-Glo results, NR-6226C treatment significantly inhibited the reduction of XTT in a concentration- and time-dependent manner (Fig. 4C), which could be due to reduced electron transport chain activity (30). We also studied

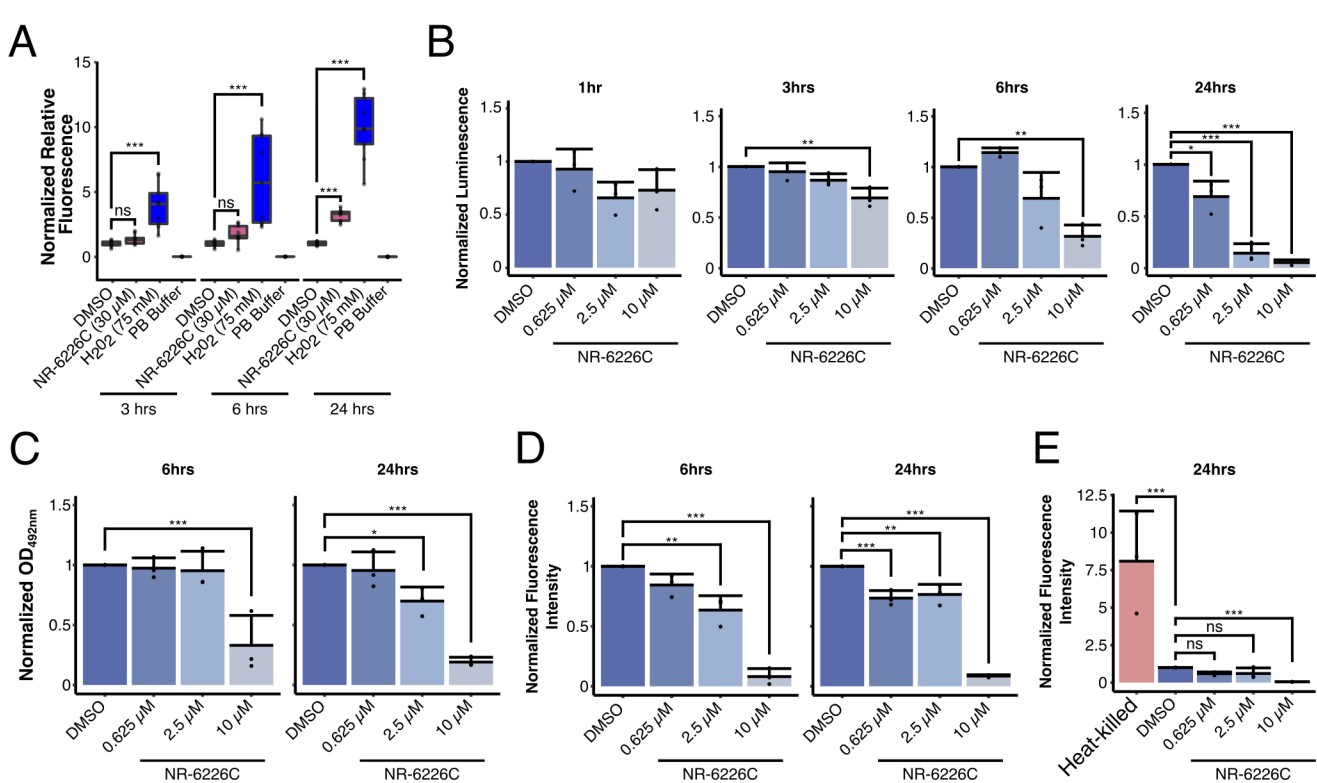

**FIG 4** NR-6226C treatment impairs mitochondrial functions and induces ROS formation. (A) *C. glabrata* cells were preloaded with the fluorescent ROS sensor $H_2$DCF-DA, after which changes in fluorescence were measured after 3, 6, and 24 h after exposure to DMSO, 30 μM NR-6226C, or 75 μM $H_2O_2$. (B) *C. glabrata* cells were treated for 1, 3, 6, and 24 h with DMSO, 0.625 μM, 2.5 μM, or 10 μM NR-6226C, after which relative ATP levels were measured by luminescence using CellTiter-Glo. An equal number of cells was harvested for each experiment, and luminescence was normalized to DMSO. (C) *C. glabrata* cells treated with DMSO, 0.625 μM, 2.5 μM, or 10 μM NR-6226C for 6 and 24 h and then treated with XTT-phenazine methosulfate. Colorimetric changes were measured using $OD_{492nm}$. An equal number of cells were harvested for each experiment, after which the $OD_{492nm}$ was normalized to DMSO. (D) *C. glabrata* cells were treated with DMSO, 0.625 μM, 2.5 μM, or 10 μM NR-6226C for 6 and 24 h, after which they were loaded with MitoTracker and measured for fluorescence using a plate reader. The autofluorescence from unstained treated samples was subtracted from the raw data and then normalized to DMSO. (E) *C. glabrata* cells were treated with DMSO, 0.625 μM, 2.5 μM, or 10 μM NR-6226C for 24 h and then stained with propidium iodide and measured for fluorescence using a plate reader. The autofluorescence from unstained samples was subtracted from the raw data and then normalized to DMSO. Statistical analyses for all panels were performed using one-way ANOVA and Tukey's HSD for post hoc analyses; ***$P \leq 0.001$; **$P \leq 0.01$; ns, not significant; $n = 3$.

mitochondria by labeling *C. glabrata* cells with the fluorescent dye MitoTracker Red CMXRos, which fluoresces preferentially upon entry into metabolically active mitochondria. Interestingly, after 6 h and 24 h treatment with NR-6226C, cells clearly exhibited lower fluorescence intensities compared to cells treated with DMSO (Fig. 4D, see Fig. S6C for examples of microscopy images), further supporting the idea that NR-6226C impairs mitochondrial activity.

The observed reduction in mitochondrial activity after treatment with NR-6226C for 24 h could be the result of loss of cell viability rather than impaired mitochondrial activity. To determine whether NR-6226C affects cell viability, we stained cells with propidium iodide, which is a membrane-impermeable nucleic acid intercalator that only stains nucleic acids in dead cells. Cells were treated with increasing concentrations of NR-6226C for 24 h, using heat-killed *C. glabrata* cells as a positive control. As shown in Fig. 4E, cells that were treated with NR-6226C showed no significant difference in fluorescence intensity compared to DMSO, showing that treatment with NR-6226C for 24 h is not sufficient to kill the fungal cells. Somewhat surprisingly, cultures treated with 10 µM NR-6226C exhibited an apparent reduction in the number of propidium iodide-positive cells compared to DMSO-treated cells (Fig. 4E). While we do not presently understand this effect, one explanation could be that high concentrations of NR-6226C, by suppressing mitochondrial functions, might reduce the rate of spontaneous, mitochondria-mediated altruistic cell death of aged cells, which normally occurs when yeast cultures reach stationary phase (31, 32). Further experiments are required to better understand this effect of NR-6226C. Nonetheless, we conclude that NR-6226C has a cytostatic effect on *C. glabrata* cells and that it causes significant mitochondrial dysfunction, leading to loss of metabolic activity and depletion of cellular ATP levels.

## NR-6226C has antifungal activity in an *in vivo* infection model

To investigate whether NR-6226C has antifungal activity *in vivo*, we employed *Galleria mellonella* as an infection model. Recently, *G. mellonella* has emerged as a robust infection model for studying virulence and antimicrobial therapies against infectious agents (33). *Galleria* is capable of mounting an efficient innate immune response against pathogenic fungi that is broadly comparable with the mammalian antifungal immune response (33). In addition to obvious ethical issues, another important advantage of *G. mellonella* over mice is that there are no concerns of introducing pathogens into breeding facilities, which is a major challenge with mouse models.

In brief, *G. mellonella* larvae were infected with WT *C. glabrata*, followed by incubation in the presence of either DMSO (untreated control) or 30 µM NR-6226C, a concentration well above the *in vitro* EC50 for fungal proliferation. As shown in Fig. 5A and B, treatment with NR-6226C significantly improved the survival of *G. mellonella* larvae infected with either wild-type *C. glabrata* or azole-resistant *C. glabrata*. We conclude that NR-6226C is a selective antifungal agent with *in vivo* activity that bypasses azole resistance.

## Synergistic antifungal effect of NR-6226C and fluconazole against *C. albicans*

One approach to countering antifungal resistance that has recently gained interest is combination therapy (34), where the simultaneous use of two drugs produces either an additive or synergistic effect to impede cellular growth. This strategy could prove beneficial in many circumstances. For example, drugs may be used at lower concentrations to inhibit fungal growth, thereby preventing general toxicity to the host. Combinations of drugs that target different fungal processes may also reduce the likelihood of development of acquired resistance to antifungal drugs. We, therefore, tested NR-6226C in combination with fluconazole, a first-line antifungal that is often clinically ineffective against common nosocomial infections, such as certain *Candida* spp. Interestingly, while wild-type *C. albicans* was only susceptible to high concentrations of fluconazole, combination of fluconazole with NR-6226C resulted in a significantly stronger impairment of fungal proliferation than either drug alone (Fig. 6A). To further determine whether the combinatorial effects of these compounds were synergistic, we analyzed our

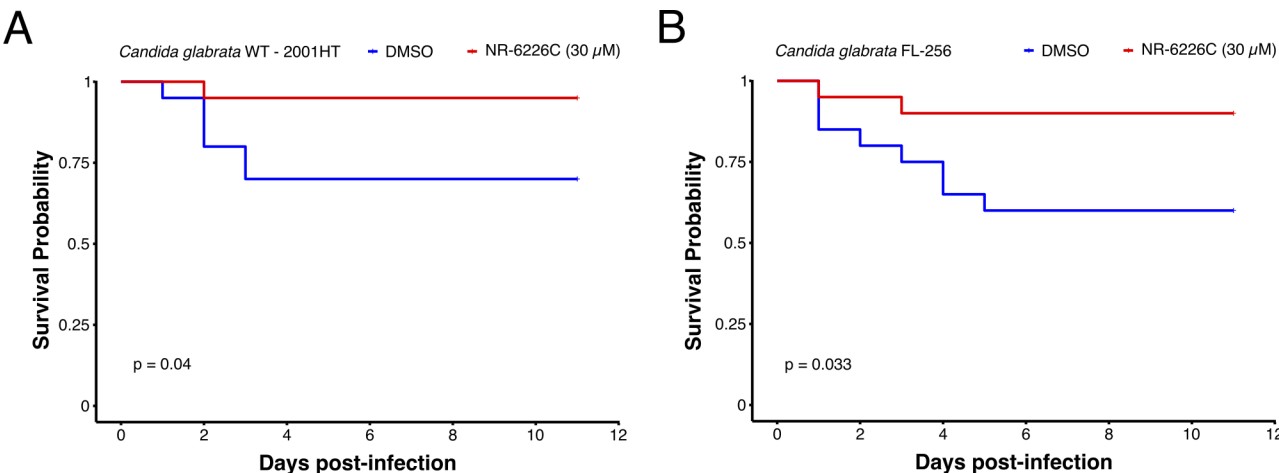

**FIG 5** NR-6226C treatment improves the survival of *C. glabrata*-infected *G. mellonella* larvae. (A and B) *G. mellonella* larvae were infected with either *C. glabrata* 2001HT (A) or *C. glabrata* FL-256 (B) and treated with either 0.15% DMSO or 30 µM NR-6226C, after which overall survival was monitored over time. Survival curves were generated using the Kaplan-Meier formula. Statistical significance was calculated using log-rank tests in R.

data using the SynergyFinder Plus package (35) for R using the Bliss model for synergistic interactions (36). This revealed strong synergy between NR-6226C and fluconazole (Fig. 6B; Fig. S7A). Although *C. glabrata* is more sensitive to NR-6226C than *C. albicans*, we did not observe substantial additive or synergistic effects of drug combinations (Fig. S7B through G). This might be expected because *C. glabrata* exhibits intrinsically low susceptibility to fluconazole (37). While analysis of combinations between NR-6226C and a broader panel of antifungal drugs against a spectrum of pathogenic fungi will be the

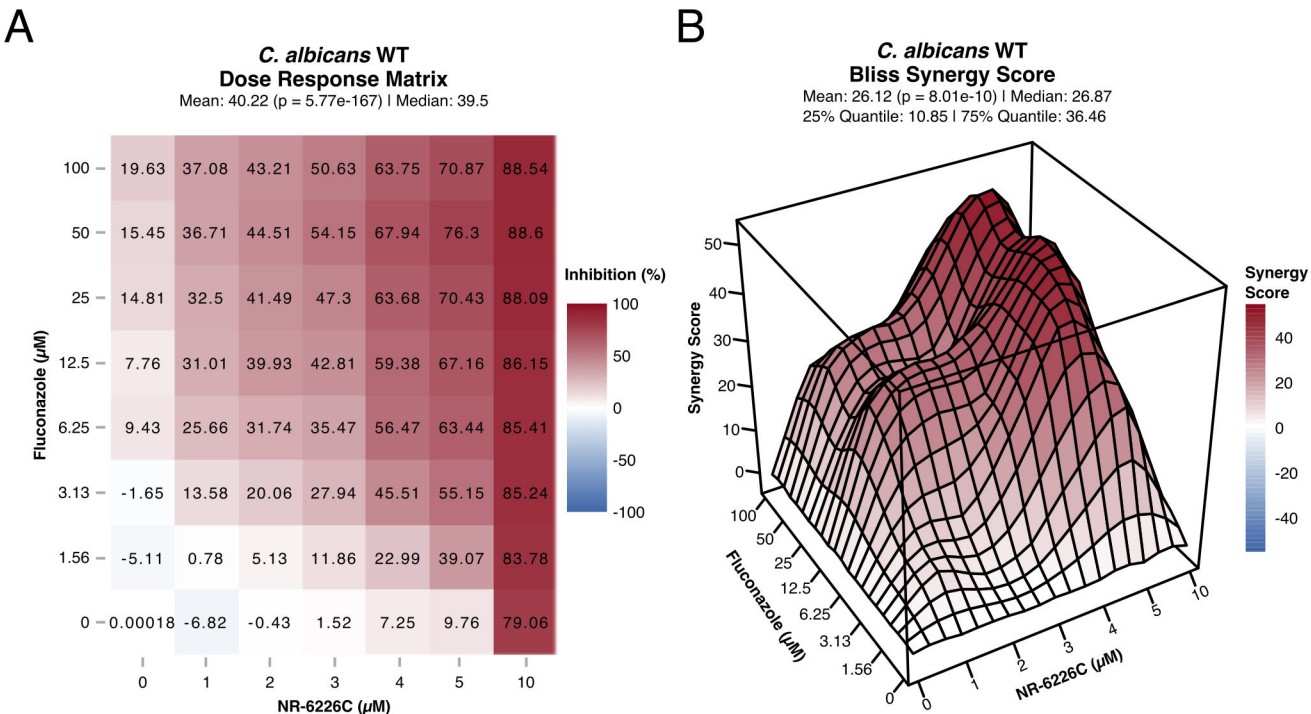

**FIG 6** Fluconazole and NR-6226C have a synergistic effect. (A) Dose-response matrix of *C. albicans* WT cells treated in combination with fluconazole and NR-6226C. Relative cell numbers were quantified as described in Fig. 1B. (B) 3D visualization of the synergy matrix shown in panel A using the Bliss synergy model. Panels A and B were generated using SynergyFinder in R.

focus of future studies, these results demonstrate that NR-6226C strongly potentiates the antifungal activity of fluconazole against *C. albicans*.

## DISCUSSION

A key challenge in combating fungal resistance is the limited selection of antifungal drugs that are clinically available. Our study revealed that the compound NR-6226C has potential as an antifungal agent. Pathogenic fungi require iron for infection, but the concentration of free iron in mammals is very low ($\sim 10^{-9}$ M) due to sequestration by iron-binding and iron-containing compounds (38). In our study, we found that NR-6226C prevented *Candida* proliferation *in vitro* and *in vivo* by chelating iron. This resulted in activation of iron starvation response genes such as *FTR1*, *HMX1*, *LSO1*, and *FET3* in an Atf1/2-dependent manner. *FET3* encodes a high-affinity iron importer, which mediates iron uptake from the extracellular environment (24), and it has been proposed that *LSO1* encodes a cytoplasmic protein involved in intracellular iron transport (16). Furthermore, to overcome the limiting iron availability in their host, pathogenic fungi increase the Aft1-dependent expression of the heme oxygenase Hmx1, which facilitates degradation of heme from the host, thereby releasing iron for subsequent metabolic use (39–41). Our results indicate that iron chelation by NR-6226C has a direct effect on fungal iron supply, thereby affecting iron metabolism and subsequent intracellular regulation.

Other than the activation of genes regulated by the Aft1 iron sensory system, our RNA sequencing data revealed the downregulation of enzymatic activities and other genes linked to mitochondrial functions. These findings are consistent with previous yeast studies showing that iron homeostasis is tightly coordinated with metabolic processes (42). Given that multiple iron-dependent metabolic processes occur in mitochondria, cells likely redirect the limited amount of available iron to essential iron-dependent pathways to maintain homeostasis. The significant decrease of the expression of Fe-S cluster-dependent genes encoding $H_2S$-*NADP* oxidoreductase, *Sdh2*, and *Aco1* (Fig. 3A), which have known functions in the mitochondrial respiratory chain (43, 44), supports this hypothesis. Further evidence that NR-6226C perturbs iron homeostasis is provided by induction of *CAGL0L11990g*, a homolog of *Sc GRX4*. Grx4 is a cytosolic monothiol glutaredoxin that not only functions in concert with Fe-S clusters as an iron sensor, but which is also involved in Fe-S cluster biogenesis in the mitochondria (45, 46). Although we have not studied the molecular mechanism by which iron depletion results in downregulation of these genes in *C. albicans*, studies in *S. cerevisiae* have revealed that this is at least in part due to Cth2-dependent post-transcriptional degradation and translational repression of these mRNAs (47, 48). It is not unlikely that similar mechanisms operate in *C. albicans*.

While NR-6226C has a strong preference for $Fe^{2+}$, the addition of $Zn^{2+}$ and $Cu^{2+}$ to the extracellular medium also countered its antifungal activity, even though these metal ions do not efficiently interact with NR-6226 *in vitro* even at high concentrations. Although we cannot exclude the possibility that the antifungal effect of NR-6226C is mediated by sequestration of trace amounts of $Zn^{2+}$ and $Cu^{2+}$ *in vivo*, we believe that it is more likely that these metals induce a stress response that compensates for iron chelation by NR-6226C. Indeed, high concentrations of heavy metals cause mismetallation (14, 49), which can induce an iron starvation response that results in increased uptake of iron from the environment by microorganisms (50).

It is well known that persistent iron deprivation leads to increased levels of oxidative stress over time (51). Similarly, long-term exposure to NR-6226C resulted in a significant increase in ROS production. However, ROS production did not appear to be the direct cause for loss of cell proliferation, as treatment with ROS scavengers did not rescue the effects of NR-6226C. Although long-term NR-6226C-induced ROS production may have a deleterious effect on fungal cells, the immediate effects of NR-6226C may rather be due to impaired metabolic activity and ATP production. These findings mirror previous studies that linked iron metabolism and mitochondrial activity to fungal pathogenicity in *C. albicans*, *C. auris*, and *Aspergillus fumigatus* (52–55). These links between mitochondria

and virulence include lipid biosynthesis, mitochondrial fusion, mitochondrial respiration, and calcium signaling and indicate that therapies affecting mitochondrial function could prove to be an effective form of antifungal therapy. Indeed, a recent study demonstrated that iron homeostasis and mitochondrial activities may be important targets for treatment of *Candida auris* (54).

An interesting finding of our study is that NR-6226C strongly synergizes with fluconazole in blocking proliferation of *C. albicans*. Drug combinations can have several advantages, such as the need for lower concentrations of drugs to achieve the desired effect, thereby reducing the risk of off-target toxicity toward host cells. Another advantage is that it is likely more difficult for cells to develop simultaneous resistance against two drugs with separate mechanisms of action. Although more comprehensive drug combination testing against a range of pathogenic fungi will be the focus of future studies, it will be particularly interesting to test combinations of NR-6226C with azoles because these antifungals exert their effect by blocking ergosterol biosynthesis and expression of genes required for ergosterol synthesis is strongly dependent on iron availability (56).

In conclusion, NR-6226C is an interesting lead compound for further development into antifungal therapy. In future studies, it will be of interest to study the effects of NR-6226C on other fungal pathogens such as *Aspergillus fumigatus* and *Cryptococcus neoformans*, and studies in mammalian models will be important to further characterize the *in vivo* potential of NR-6226C, and further studies may be needed to optimize its antifungal activity and its absorption, distribution, metabolism, and excretion profile.

## MATERIALS AND METHODS

### Culture maintenance

*Candida* spp. and *S. cerevisiae* strains used in this study are listed in Table S1. Cells were stored at −80°C in YPD (1% yeast extract, 2% peptone, and 2% dextrose) media containing 20% glycerol. For experimental use, cells were streaked out on solid YPD plates (with 2% agar) and incubated at 30°C for 2–3 days until single colonies appeared. The solid plates were stored at 4°C for up to 3 weeks until new cells were streaked out again.

For experiments, yeast cells were routinely cultured in YPD media and incubated at 30°C overnight. The cultures were refreshed the next day for 2–3 h in YNB −iron (0.69% yeast nitrogen base without amino acids or iron, with 2% dextrose) media, unless stated, at the same temperature. After incubation, the optical density ($OD_{600}$) of the cell cultures was measured and adjusted to 0.15.

### Compound screening

The collismycin compounds were provided by EntreChem (Spain) at stock solutions of 20 mM in DMSO and stored at −20°C. For serial dilutions of the compounds in clear 96-well plates, intermediate stock concentrations of 100 µM of compound (100 µL for each technical replicate) were prepared in YNB −iron media unless otherwise specified, and 1:1 serial dilutions were performed using a multi-channel pipette. Fifty microliters of cell culture ($OD_{600} = 0.15$) were added to the wells containing the compounds, for a total volume of 150 µL. The final concentrations in the wells (with three technical replicates) were equivalent to 0.5, 1, 2.1, 4.2, 8.4, 16.7, 33.3, and 66.6 µM. The $OD_{600}$ of each well was quantified, and the plates were incubated inside plastic bags (to prevent evaporation) at 30°C for 24 h, after which the final $OD_{600}$ was measured. The $EC_{50}$ was calculated using the four-parameter log logistic function with the drc package in R (57).

## Human cell lines

HEK-293 and HS-5 cells were stored in liquid nitrogen, in freezing medium [50% Dulbecco's Modified Eagle's Medium (DMEM), 40% fetal bovine serum, and 10% DMSO] for long-term storage. For experimental use, the cells were thawed and then cultured using DMEM containing 10% fetal serum albumin and 1% penicillin/streptomycin.

## CellTiter-Glo assay for human cells

The 1:1 serial dilutions of the compounds were performed in 96-well plates to obtain a final concentration range (three technical replicates each) equivalent to 0.781, 1.562, 3.125, 6.25, 12.5, 25, 50, and 100 µM. Each well had a total volume of 100 µL, consisting of 50-µL compound and 50-µL seeded cells (10,000 cells per well). The plates were placed inside plastic bags to prevent evaporation and incubated at 37°C, 5% $CO_2$ for 24 h. One hundred microliters of CellTiter-Glo 2.0 (Promega, G9243) was added to each well and then measured for luminescence using a Synergy 2 Gen 5 plate reader from BioTek.

## CellTiter-Glo assay in *Candida*

A volume of 0.5-mL cell culture in YNB −iron ($OD_{600}$ = 0.15) was mixed to a final volume of 1 mL in the following treatments: 0.05% DMSO, 0.625 µM NR-6226C, 2.5 µM NR-6226C, or 10 µM NR-6226C. The samples were incubated for 24 h at 30°C and then measured for $OD_{600}$. Equal number of cells were harvested in each time point according to an $OD_{600}$ of 0.1–0.4 and then centrifuged at 7000 rpm for 5 minutes to remove the supernatant. The cell pellets were resuspended in 100-µL YNB −iron and then transferred to a 96-well plate. Finally, the samples were treated with 100-µL CellTiter-Glo and then measured for luminescence using a plate reader.

## Nano liquid chromatography-mass spectrometry

Liquid chromatography-mass spectrometry (LC-MS)-grade water, acetonitrile, and formic acid (≥99%) were purchased from VWR (Radnor, PA, USA). NR-6226C [terpyridine (2,2′:6′,2′′-terpyridine, 98%)] was purchased from Sigma-Aldrich (now Merck, KGaA, Darmstadt, Germany). Metal solutions containing 2 µM, 4 µM, 6 µM, and 10 µM $CaCl_2$, $CuCl_2$, $FeCl_2$, $FeCl_3$, $MgCl_2$, $MnCl_2$, and $ZnCl_2$ in 0.1% formic acid in LC-MS-grade water (hereby referred to as 0.1% formic acid) were made. A stock solution of terpyridine (100 µM) in 0.1% formic acid was made and added to the metal solutions to a final concentration of 10 µM terpyridine. All metal solutions in addition to a blank sample containing 10 µM terpyridine in 0.1% formic acid were subjected to nano liquid chromatography (nanoLC-MS) analysis. nanoLC-MS was performed using an nLC EASY 1000 pump connected to a Q-Exactive mass spectrometer (MS) with a Nanospray Flex ion source (all from Thermo Fisher Scientific, Waltham, MA, USA). For separation, a 50 µm (inner diameter) × 5 cm in-house packed nanoLC column containing 2.6-µm Accucore C18 particles (80 Å) was used. The column was packed according to the protocol from Berg et al. (58). Trapping of analytes prior to separation was performed online using a 2-cm Acclaim PepMap with 3-µm particles (100 Å). All columns and packing materials were obtained from Thermo Fisher Scientific. The mobile phases consisted of 0.1% formic acid (A) and 90/10/0.1 acetonitrile/water/formic acid (vol/vol/vol) (B). A linear gradient from 5% to 20% B in 10 minutes was employed, and trapping was performed by 100% mobile phase A. Equilibration was performed using 2-µL and 3-µL 100% A for the trapping column and analytical column, respectively. The flow rate was set to 250 nL/min, and the injection volume used was 1 µL.

The MS was operated in positive mode using single ion monitoring. A mass-to-charge ratio (*m/z*) of 234.1 (M + H) was used for terpyridine, and an *m/z* of 261.1 $(2M + Fe)^{2+}$ was used for terpyridine bound to $Fe^{2+}$. All data processing was performed by using the XCalibur Software (Thermo Fisher Scientific).

## Cell proliferation with metal supplementation

NR-6226C (Sigma-Aldrich) was prepared at a stock solution of 20 mM. One-millimolar stocks of metal solutions ($CaCl_2$, $CuCl_2$, $FeCl_2$, $FeCl_3$, $MgCl_2$, $MnCl_2$, and $ZnCl_2$) were prepared in $mQH_2O$ and autoclaved. Serial dilutions of NR-6226C using YNB −iron media were performed in clear 96-well plates to obtain final concentrations of 0.78, 1.56, 3.13, 6.25, 12.5, 25, 50, and 100 µM, after the addition of metals and cells. Fifty microliters of either $mQH_2O$ or metal solution were added to each well, including 50 µL of refreshed *C. glabrata* WT with an adjusted $OD_{600}$ of 0.15. The 96-well plates were measured at $OD_{600}$ and incubated in plastic bags to prevent evaporation for 24 h, after which the final $OD_{600}$ was measured.

## RNA sequencing

Cells from overnight precultures in YPD medium were reinoculated in fresh YNB −iron medium (see "Culture maintenance" for contents) without iron and grown until $OD_{600}$ of 0.4. Cultures were then treated for 1 h with DMSO (vehicle) or 10 µM NR-6226C or 5 µM Dp44mT. Cells were collected from three independent experiments, snap frozen, and stored at −80°C. Total RNA purification was performed as previously described using the RNeasy Mini Kit (74104, Qiagen), and gDNA removal was made using the RNase-Free DNase Set kit (79254, Qiagen). RNA quantification and quality control were done with a Tape Station 4150 (Agilent) (59).

Library preparation and sequencing were performed at GenomEast (IGBMC, Illkirch, France). One DMSO 0.05% sample was excluded from the analysis due to insufficient quality. The library was sequenced on Illumina Hiseq 4000 sequencer as single-read 50-base reads following Illumina's instructions. Reads were pre-processed in order to remove adapter, polyA, and low-quality sequences (Phred quality score below 20). Reads shorter than 40 bases were discarded from further analysis. These pre-processing steps were performed using cutadapt version 1.10 (60). Image analysis and base calling were performed using RTA 2.7.7 and bcl2fastq2.17.1.14. Adapter dimer reads were removed using DimerRemover (https://sourceforge.net/projects/dimerremover/). The quality of the RNAseq reads was examined using FastQC 0.11.2 (http://www.bioinformatics.babraham.ac.uk/projects/fastqc/) and FastQScreen 0.5.1 (https://www.bioinformatics.babraham.ac.uk/projects/fastq_screen/). Reads were mapped onto the ASM254v2 assembly of *Candida glabrata* genome using STAR version 2.5.3a (61). Gene expression quantification was performed from uniquely aligned reads using htseq-count version 0.6.1p1 (62), with annotations from Ensembl fungi version 50 and union mode. Only non-ambiguously assigned reads have been retained for further analyses. Read counts were normalized across samples with the median-of-ratios method (63). Comparisons were implemented in the Bioconductor package DESeq2 version 1.16.1 using the test for differential expression (64). Genes with no *P*-value corresponded to genes with high Cook's distance that were filtered out. *P*-values were adjusted for multiple testing using the Benjamini and Hochberg method (65). Genes with no adjusted *P*-value correspond to genes filtered out in the independent filtering step in order to remove reads from genes that have no or little chance of showing evidence of significant differential expression.

## RT-qPCR

WT and *atf1Δ* cells were grown overnight in complete supplement mixture (CSM). Cells were then diluted 10 times and grown until log phase at which point 10 µM of NR6226C or Dp44mt was added to the medium. After 24 h of incubation, cells were collected, and total RNA purification, reverse transcription, and RT-qPCR were performed as previously described with minor modifications (59, 66). Briefly, total RNA was purified using the RNeasy Mini Kit (74104, Qiagen), and reverse transcription was performed using the QuantiTect Reverse Transcription Kit (205311, Qiagen). RT-qPCR experiments were done using the HOT FIREPol EvaGreen qPCR Mix (08-36-00001-10, Solis Biodyne) and a LightCycler 96 System (Roche). Expression of the 5S gene was used as a control. The primers used in RT-qPCR experiments are as follows:

*FTR1*: GATTGGGTTCTTGAGTAGAAG and GAGCCCTGTGTGGTAATA;
*FET3*: GTCAATATGAAGACGGGATG and CCACTCACTAAGCGATAAAG;
*HMX1*: CCAGAGATGCCCACAATA and CATAATAGTACGCCAGAATACC;
*LSO1*: AGAAGAAGCTGATTATGGAAC and CTGCCTCCCTTACCTAAA; and
*5S:* GTTGCGGCCATATCTACCAGAAAG and CGTATGGTCACCCACTACACTACT.

## ROS assay

Our method was adapted using the protocol developed by James et al. (67) for ROS assessment in yeast. The overnight culture was refreshed in YNB −iron media for 2–3 h and then adjusted to $OD_{600}$ of 0.5 in 100 mL. The cells were washed by centrifugation for 5 minutes at 4,300 rpm, followed by aspiration of the supernatant and a resuspension of the cell pellets in 20-mL phosphate buffer (PB) solution (0.1 M, pH 7.4). One milliliter of cell suspension was aliquoted to Falcon tubes and then centrifuged again to remove the supernatant. The general oxidative stress indicator, CM-$H_2$DCFDA (Thermo Fisher), was prepared in a stock solution of 1 mM DMSO and then diluted to 10 µM using PB solution. The cell pellets were pre-loaded with the dye by resuspension in 10 µM CM-$H_2$DCFDA and then incubated in the dark at 30°C for 30 minutes. Cell pellets that were used for a negative control of the dye were resuspended in PB solution alone. After incubation, the samples were centrifuged at 4,300 rpm for 2 minutes and followed by aspiration of the supernatant. The cell pellets were resuspended in YNB −iron and were then treated with 0.15% DMSO, 30 µM NR-6226C, or 75 mM hydrogen peroxide ($H_2O_2$), including the negative controls. The samples were placed in a 30°C incubator and then measured for $OD_{600}$ after 3, 6, and 24 h of incubation, after which an equal number of cells ($OD_{600}$ = 0.5, 1 mL) were harvested. The harvested cells were centrifuged at $6,000 \times g$ for 10 minutes, with the supernatant aspirated and the cells resuspended in PB solution. The fluorescence of the samples was measured in a black, clear 96-well plate using a spectrofluorometer.

## XTT metabolic assay

XTT (Invitrogen) and phenazine methosulfate (Sigma) were freshly prepared to 1 mg/mL in phosphate-buffered saline (PBS) and 320 µg/mL in milliQ $H_2O$ for every experiment, respectively. XTT was sterilized using a 0.2-µm polyethersulfone syringe filter prior to use and then mixed with phenazine methosulfate in a ratio of 10:1 (XTT:phenazine methosulfate). Refreshed cultures of *C. glabrata* WT (10028) cells with $OD_{600}$ of 0.6 were centrifuged at 7,000 rpm for 5 minutes to obtain cell pellets. Subsequently, cell pellets were resuspended in 6-mL 0.05% DMSO, 2.5 µM, or 10 µM NR-6226C and then incubated at 30°C. After 6 and 24 h, the $OD_{600}$ of the treated cell cultures were measured, and equal numbers of cells were harvested according to an $OD_{600}$ of 0.3 in 2 mL. The harvested cells were centrifuged at 7,000 rpm for 5 minutes, followed by aspiration of the media and the re-suspension of cell pellets in 100 µL of PBS. The cells were transferred to a 96-well plate and added with 100 µL of XTT-phenazine methosulfate solution. The plate was incubated for 1 h at 37°C and then measured for absorbance at 492 nm using a microplate reader (29, 68).

## Mitochondrial staining with MitoTracker

MitoTracker Red CMXROS (Invitrogen) was prepared to a stock solution of 1 mM using DMSO and stored at −20°C. Five-milliliter *C. glabrata* WT (JEY10028) cells with $OD_{600}$ of 0.4 were centrifuged at 7,000 rpm for 5 minutes to obtain cell pellets. The cell pellets were resuspended in 5-mL 0.05% DMSO, 2.5 µM, or 10 µM NR-6226C and then incubated at 30°C for the amount of time indicated in the figures. The samples were measured for $OD_{600}$, followed by harvesting of a total number of cells equivalent to a 500-µL culture with $OD_{600}$ of 0.4 and then centrifuged at 7,000 rpm for 5 minutes. The cell pellets were resuspended in 100 nM MitoTracker (in milliQ $H_2O$) and incubated at 37°C for 25 minutes. The samples were centrifuged at 7,000 rpm for 5 minutes, and the resulting pellets were

washed with 1-mL milliQ $H_2O$. The cell pellets were resuspended in milliQ $H_2O$ and then imaged using a Zeiss LSM 710 microscope.

To quantify the fluorescence intensity of the cells, separate experiments were performed using equal numbers of cells ($OD_{600nm} = 0.4$ in 5 mL) and staining concentration as described above. Cells stained with MitoTracker were resuspended in 100 μL of milliQ $H_2O$ and then transferred to flat, black 96-well plates. The fluorescence intensity was measured using a BioTek plate reader (69). The autofluorescence of the unstained treated samples was subtracted from the raw fluorescence intensities of the stained samples and then normalized to DMSO. Normalized values were analyzed using one-way analysis of variance (ANOVA) and Tukey's honest significant difference (HSD) post hoc test.

## Dead cell staining with propidium iodide

Propidium iodide (Sigma) was freshly prepared to a final concentration of 5 μg/mL in PBS.

Cells adjusted to an $OD_{600}$ of 0.6 were centrifuged at 7,000 rpm for 5 minutes, followed by resuspension in 5-mL 0.05% DMSO, 0.625 μM, 2.5 μM, or 10 μM NR-6226C. The samples were incubated at 30°C and harvested to a total number of cells equivalent to a 5-mL culture with an $OD_{600}$ of 0.3 at 6 and 24 h. As a positive control for cell death, cells treated with DMSO were incubated at 75°C for 1 h. The cells were centrifuged at 7,000 rpm for 5 minutes to remove the supernatant, followed by resuspension of the cell pellets in 100-μL 5 μg/mL propidium iodide for 10 minutes in the dark at room temperature. The supernatant was removed by centrifugation at 7,000 rpm for 5 minutes, followed by washing with 500-μL PBS and a final centrifugation at 7,000 rpm for 5 minutes. Finally, the cells were resuspended in 150-μL PBS and measured for fluorescence intensity using a microplate reader.

The autofluorescence of the unstained treated samples was subtracted from raw fluorescence intensities of the stained samples and then normalized to DMSO. Normalized values were log transformed and then analyzed using one-way ANOVA and Tukey's HSD post hoc test.

## Survival assays in infected *Galleria mellonella* models

*Galleria mellonella* was obtained from R.J. Mous Livebait (The Netherlands). The larvae were selected by weight (0.2–0.3 grams) and by the absence of dark spots on the cuticle. The larvae were maintained at room temperature, and the day before the experiment, they were transferred to the temperature at which the experiment was going to be performed (30°C). The number of dead caterpillars was scored every day. A group of 20 larvae were incubated with PBS and 0.15% DMSO and with NR-6226C alone as controls in every experiment. *G. mellonella* were infected with either *C. glabrata* 2001HT or FL-256; *C. glabrata* cells were grown in liquid media (Oxoid) overnight at 30°C with moderate shaking (150 rpm). Groups of 20 larvae per strain were infected in each experiment. The pro-leg area was cleaned with 70% ethanol using a swab. The larvae were inoculated with 10 μL of *C. glabrata* suspension at $2.5 \times 10^6$ cells/mL in PBS containing 50 μg/mL of ampicillin by injection, using a Hamilton syringe with 26-gauge needle in the last left pro-leg. Within 2 h of infection, 10 μL of NR-6226C (30 μM) in PBS was injected into a different pro-leg using the same technique.

## Combination treatment with fluconazole and NR-6226C

Stock solutions of fluconazole (TargetMol) and NR-6226C (20 mM in DMSO) were prepared. Fluconazole (1.56, 3.12, 6.25, 12.5, 25, 50, and 100 μM) and NR-6226C (1, 2, 3, 4, 5, and 10 μM) were dispensed into clear 96-well plates using CERTUS FLEX liquid dispenser. Fifty microliters of refreshed *Candida* cells equivalent to $OD_{600}$ of 0.15 in YNB −iron were added to the wells and measured for $OD_{600}$. The plates were placed in plastic bags and incubated at 30°C for 24 h until the final $OD_{600}$ was measured again. Cell

inhibition was quantified by normalizing cell proliferation ($OD_{600}$) to DMSO controls and then subtracting the resulting values from 1.

## ACKNOWLEDGMENTS

This work was supported by grants from the Norwegian Cancer Society (project numbers 182524 and 208012), the Norwegian Health Authority South-East (2017064, 2017072, 2018012, and 2019096), the Research Council of Norway (261936, 301268, and 262652). O.Z. is supported by grant PID2020-114546RB by MCIN/AEI/10.13039/501100011033.

Sequencing was performed by the GenomEast platform, a member of the "France Genomique" consortium (ANR-10-INBS-0009).

We thank Cecilie Torp Andersen (Oslo University Hospital) for providing patient-derived *Candida* strains that were used in this work. We also thank all members of the Enserink group and the Knævelsrud group for helpful discussions and feedback.

## AUTHOR AFFILIATIONS

[1]Department of Molecular Cell Biology, Institute for Cancer Research, The Norwegian Radium Hospital, Montebello, Norway

[2]Centre for Cancer Cell Reprogramming, Institute of Clinical Medicine, Faculty of Medicine, University of Oslo, Oslo, Norway

[3]Section for Biochemistry and Molecular Biology, Faculty of Mathematics and Natural Sciences, University of Oslo, Oslo, Norway

[4]EntreChem SL, Vivero Ciencias de la Salud, Calle Colegio Santo Domingo Guzmán, Oviedo, Spain

[5]Mycology Reference Laboratory, National Centre for Microbiology, Instituto de Salud Carlos III, Carretera Majadahonda-Pozuelo, Madrid, Spain

[6]Department of Chemistry, University of Oslo, Oslo, Norway

[7]Center for Biomedical Research in Network in Infectious Diseases, CB21/13/00105, Instituto de Salud Carlos III, Madrid, Spain

[8]Department of Bacteriology, Norwegian Institute of Public Health, Oslo, Norway

## PRESENT ADDRESS

Javier González-Sabín, Fusoni Componentes SL, Polígono de Argame, C. Mostaya, Spain

Nicolás Ríos-Lombardía, Laboratorio de Química Sintética Sostenible (QuimSinSos), Departamento de Química Orgánica e Inorgánica, (IUQOEM), Centro de Innovación en QuímicaAvanzada (ORFEO-CINQA), Facultad de Química, Universidad de Oviedo, Oviedo, Spain

## AUTHOR ORCIDs

Oscar Zaragoza  http://orcid.org/0000-0002-1581-0845
Ignacio Garcia  http://orcid.org/0000-0002-0758-1894
Jorrit M. Enserink  http://orcid.org/0000-0002-2394-5387

## FUNDING

| Funder | Grant(s) | Author(s) |
| --- | --- | --- |
| Kreftforeningen (NCS) | 182524, 208012 | Jorrit M. Enserink |
| Ministry of Health and Care Services \| Helse Sør-Øst RHF (sorost) | 2017064, 2018012, 2019096 | Jorrit M. Enserink |
| Ministry of Health and Care Services \| Helse Sør-Øst RHF (sorost) | 2017072 | Ignacio Garcia |
| Norges Forskningsråd (Forskningsrådet) | 261936, 301268, 262652 | Jorrit M. Enserink |

| Funder | Grant(s) | Author(s) |
|---|---|---|
| Ministerio de Ciencia e Innovación (MCIN) | PID2020-114546RB | Oscar Zaragoza |

## AUTHOR CONTRIBUTIONS

Jeanne Corrales, Conceptualization, Data curation, Formal analysis, Investigation, Methodology, Validation, Visualization, Writing – original draft, Writing – review and editing | Lucia Ramos-Alonso, Data curation, Formal analysis, Investigation, Methodology, Visualization | Javier González-Sabín, Methodology, Resources | Nicolás Ríos-Lombardía, Methodology, Resources | Nuria Trevijano-Contador, Formal analysis, Investigation, Methodology, Resources, Validation, Visualization, Writing – original draft | Henriette Engen Berg, Formal analysis, Investigation, Methodology, Visualization | Frøydis Sved Skottvoll, Formal analysis, Investigation, Methodology, Visualization | Francisco Moris, Methodology, Resources | Oscar Zaragoza, Formal analysis, Funding acquisition, Investigation, Methodology, Resources, Visualization, Writing – original draft | Pierre Chymkowitch, Formal analysis, Funding acquisition, Investigation, Methodology, Resources, Supervision, Visualization, Writing – original draft, Writing – review and editing | Ignacio Garcia, Conceptualization, Data curation, Formal analysis, Funding acquisition, Investigation, Methodology, Project administration, Resources, Supervision, Visualization, Writing – original draft, Writing – review and editing | Jorrit M. Enserink, Conceptualization, Formal analysis, Funding acquisition, Investigation, Methodology, Project administration, Supervision, Writing – original draft, Writing – review and editing

## DATA AVAILABILITY

The RNAseq data set has been deposited in the ENA database under accession number PRJEB63373.

## ADDITIONAL FILES

The following material is available online.

### Supplemental Material

**Fig. S1 (Spectrum02594-23-s0001.pdf).** Supporting figure.
**Fig. S2 (Spectrum02594-23-s0002.pdf).** Supporting figure.
**Fig. S3 (Spectrum02594-23-s0003.pdf).** Supporting figure.
**Fig. S4 (Spectrum02594-23-s0004.pdf).** Supporting figure.
**Fig. S5 (Spectrum02594-23-s0005.pdf).** Supporting figure.
**Fig S6 (Spectrum02594-23-s0006.pdf).** Supporting figure.
**Fig. S7 (Spectrum02594-23-s0007.pdf).** Supporting figure.
**Table S1 (Spectrum02594-23-s0008.docx).** Strain list.
**Table S2 (Spectrum02594-23-s0009.xlsx).** Gene regulation.

### Open Peer Review

**PEER REVIEW HISTORY (review-history.pdf).** An accounting of the reviewer comments and feedback.

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
