## [Reviewer comments · Microbiology Spectrum]

Microbiology Spectrum

Characterization of a selective, iron-chelating antifungal compound that disrupts fungal metabolism and synergizes with fluconazole

Jorrit Enserink, Javier González-Sabín, Nicolás Ríos-Lombardía, Francisco Moris, Nuria Trevijano-Contador, Oscar Zaragoza, Henriette Engen Berg, Frøydis Sved Skottvoll, Lucia Ramos-Alonso, Pierre Chymkowitz, Ignacio Garcia, and Jeanne Corrales

Corresponding Author(s): Jorrit Enserink, Oslo Universitetssykehus

Review Timeline:

Submission Date:	July 7, 2023
Editorial Decision:	August 1, 2023
Revision Received:	September 23, 2023
Editorial Decision:	October 13, 2023
Revision Received:	November 27, 2023
Accepted:	December 6, 2023

Editor: Renato Kovacs

Reviewer(s): The reviewers have opted to remain anonymous.

Transaction Report:

DOI: <https://doi.org/10.1128/spectrum.02594-23>

August 1, 2023

Prof. Jorrit Enserink
Oslo Universitetssykehus
Ullernchausseen 70
Oslo
Norway

Re: Spectrum02594-23 (Characterization of a selective, iron-chelating antifungal compound that disrupts fungal metabolism and synergizes with fluconazole)

Dear Prof. Jorrit Enserink:

Link Not Available

Sincerely,

Renato Kovacs

Journals Department
Reviewer comments:

Reviewer #1 (Comments for the Author):

In this manuscript, the authors identify NR-6226C, a structural analog of collismycin A, as capable of inhibiting the growth of *Candida albicans* and *Candida glabrata*. The authors go on to characterize its effect on *C. glabrata* physiology through RNA-seq, making a case that iron chelation is the mechanism underlying its antifungal activity. Additionally, the authors show a reduction in mortality in a *Galleria* model of *C. glabrata* virulence, and that NR-6226C synergizes with fluconazole using a Bliss model for synergy.

Comments and suggestions:

Lines 110-113: You write "the clinical isolate FL-256 112 showing the greatest sensitivity." The heat maps in Figure 1b do not

appear to support this (WT *C. glabrata* appears to be the most sensitive). Going into figure S3 to see the raw data used to calculate the EC50 and generate the heat map, there appears to be wide variability between biological replicates for FL-256. Were statistical methodologies employed that accounted for the variability in these data when calculating the EC50 for FL-256 112?

Line 133-135: You state "NR-6226C and NR-6226K have antifungal activity exceeding that of ColA, with relatively low toxicity against mammalian cells." Can that be said of NR-6226K, given that you have reason to believe it was insoluble in your toxicity assays, and is likely not bioavailable?

Figure S2 is probably extraneous. Much of what it contains is negative data, and the data it contains from the hits from the compound screen are duplicated elsewhere in the manuscript. Similarly, it seems that Figure 1B and 1C represent data from the same set of experiments; if so, they could be condensed into one to let the reader know comparisons across them can be made. The data in Figure S6 are important to making the case that NR-6226C induces an iron starvation response. I would consider moving it from the supplement into the body of the manuscript.

Conventionally, effective antimicrobial concentrations and synergy are discussed in the literature in terms of MIC and FIC. EC50, in this case, is probably equivalent to MIC50, but the authors use a Bliss model for synergy in this manuscript. Is this an appropriate synergy test for antimicrobials, where the effective concentration is determined by inhibition of the growth of a whole organism, rather than inhibition of a molecular target?

Minor comments and suggestions:

Lines 36 and 41: Azoles and echinocandins are antifungals, not antibiotics. Consider "antimicrobials," as antibiotics conventionally refers to antibacterial compounds.

Line 50: the data presented do not differentiate between fungicidal activity and fungistatic activity. "Killing" is probably inappropriate here. Consider "inhibition of."

Sentence beginning on line 212: Suggest changing "*Candida* sp." to *Candida glabrata*.

Consider clarifying in the subheading on line 259 that synergy between NR 6226C and fluconazole exists for *C. albicans*, and not *C. glabrata*.

It would be great if Table S2 also included the fold changes of those altered transcripts, but that is a wishlist item.

Reviewer #2 (Comments for the Author):

The paper describes the antifungal effects of a lead compound that shows promise as a potential new antifungal. Collismycin A derivatives were screened and one compound, NR-6226C, inhibited the proliferation of WT *C. albicans* and *C. glabrata*, and two antifungal-resistant *C. glabrata* strains, while the proliferation of two human cell lines was much less impaired, suggesting there may be a therapeutic window for the compound. The authors hypothesize that NR-6226C exerts its antifungal effects through iron chelation, inducing an iron starvation response and inhibiting mitochondrial function, although the experimental evidence for this mechanism is less robust. Using a *G. mellonella* model, the authors demonstrate that NR-6226C has antifungal effects in vivo, significantly improving the survival of larvae infected with either WT or azole-resistant *C. glabrata*, although at a much higher concentration of NR-6226C than that required to inhibit fungal proliferation in vitro. Finally, NR-6226C and fluconazole have a synergistic effect against *C. albicans* in vitro, although this synergy is not seen in *C. glabrata*, which the authors hypothesize may be due to *C. glabrata*'s low susceptibility to fluconazole. Overall, the paper describes an intriguing potential new antifungal compound; the text could benefit from some minor edits for clarity. The degree of certainty regarding the compound's mechanism of action may be somewhat overstated and is arguably the paper's greatest shortcoming; however, the authors' findings appear to be experimentally sound overall, and the results still represent a useful contribution to the field.

The following points are suggestions to improve the clarity of the paper.

1. Antibiotic vs antimicrobial, antifungal: In several places throughout the paper (such as lines 36, 41, 85), the term "antibiotic" is used to refer to a compound with antifungal properties; as "antibiotic" is most commonly used to mean "antibacterial," the term "antimicrobial" might be more appropriate.
2. Lines 116-118: I think the authors mean either an EC50 12x smaller (not greater) than, or an inhibitory effect 12x greater than ColA. It would be helpful to give the range of EC50s in the text (1.00 +/- 0.49 to 4.78 +/- 0.53) to make it easier for the reader to compare these values to those given for human cell lines in the next paragraph.
3. Line 156: Do the authors have data showing that NR-6226C binds Fe³⁺ in a 2:1 complex? I would suggest either including it in Supplemental Fig 5 or saying "data not shown." Or perhaps the authors are referring to the data shown in Fig 2B and simply mean that it binds to Fe³⁺? It's a very minor point which the authors may choose to ignore, I was simply confused about whether or not they meant to indicate the binding ratio was the same.
4. The authors could consider softening the language when stating their conclusions about NR-6226C being an iron chelator (for example, on lines 149 and 166, could change "showing" and "indicate" to "suggesting"/"suggest"). While their hypothesis is plausible, at least in 2D, the spacing of the three putative iron-binding N atoms in NR-6226C looks rather different from that in ColA (2 vs 3 carbons between N atoms); the lack of the third N is not the only structural difference between ColA and ColH; and the fact that CuCl₂ attenuates the effects of 26C as effectively as FeCl₂ (and ZnCl₂ as effectively as FeCl₃) may indeed be due to mismetallation, but it is also possible that 26C acts via a mechanism other than iron binding.
5. Treatment with NR-6226C induces an iron starvation response: was there a particular reason for choosing the experimental conditions for the mRNA expression experiments? The authors explain their reason for choosing 1 h (as opposed to 24 h) for treating the *C. glabrata* cells, but not how they chose 10 μ M 26C (quite a bit higher than the EC50). I assumed that 5 μ M

Dp44mT was chosen because it was a concentration that was known to be effective. However, they then treated with *S. cerevisiae* for 24 h with 10 μ M 26C or 10 μ M Dp44mT. If there was a particular rationale for this choice, it might be helpful to clarify that in the text.

6. The authors might wish to say something about Fig 3B in the text. In line 191, I believe they are referring to Fig 3C.

7. Line 211: I believe this should say Fig 3D, not Fig 3C.

8. Treatment with NR-6226C induces ROS formation: it would be helpful for the authors to explain why they chose to use 30 μ M 26C, which is much higher than either the EC50 or the concentration used in the gene expression experiments. It seems a bit odd that a transcriptional response to oxidative damage could be detected after 1h treatment with 10 μ M 26C, but that no significant change in ROS formation was detectable after 3 or 6 h treatment with 30 μ M 26C, unless the cells are mounting such an effective response that the ROS production is not detectable, or the cells have simply died. According to supplemental fig S7A (not referred to in the text), it is clear that there is little, if any, proliferation occurring in cells treated with 26C or H₂O₂; are the cells still viable? For the experiments shown in Figs 4B and 4C, it is stated that an equal number of cells was harvested under each condition, yet in Fig 4C there appear to be far fewer cells in the 10 μ M 26C condition than the 2.5 μ M 26C condition. If the cells treated with 30 μ M 26C mostly died, that could have contributed to the difficulty in detecting ROS formation in Fig 4A. Similarly, it is unclear whether the data in Fig 4B represent a loss of mitochondrial activity and ATP production specifically, or whether this simply demonstrates that the cells are no longer viable (for reasons that may or may have been caused by mitochondrial dysfunction). Regardless of whether or not the cells survived, to support a mechanistic argument, it would have been more convincing to see an effect on cellular function at a concentration and timepoint at or below the EC50 and 24 hours. Negative data (no measurable effect under conditions that inhibit proliferation) are still important, but would have supported a different argument.

9. In vivo infection model: It is understandable that a higher concentration may be required in an animal model, but it might be useful to acknowledge that the authors chose to use a higher concentration of 26C in this model than was needed to inhibit proliferation in vitro.

10. Lines 311-312: The mechanism of downregulation in response to iron depletion may indeed be similar in *C. albicans*, but I'm not sure I would say that there is a high degree of conservation between *S. cerevisiae* and *C. albicans*.

11. Figure 2B: I believe there is an "l" missing in the word "metal" in the y-axis label.

12. Figure 3B: what is the difference between the two lines labeled "Function: Methyltransferase activity"? They both say "n=8," but have slightly different ratios.

13. Other minor suggestions for clarity: please see suggested changes in Word document. In particular, there are a number of small but important suggested clarifications to the methods section.

Staff Comments:

Preparing Revision Guidelines

Please return the manuscript within 60 days; if you cannot complete the modification within this time period, please contact me. If you do not wish to modify the manuscript and prefer to submit it to another journal, please notify me of your decision immediately so that the manuscript may be formally withdrawn from consideration by Microbiology Spectrum.

Characterization of a selective, iron-chelating antifungal compound that disrupts fungal metabolism and synergizes with fluconazole

Jeanne Corrales^{1,2,3}, Lucía Ramos-Alonso³, Javier González-Sabín^{4,5}, Nicolás Ríos-Lombardía^{4,6}, Nuria Trevijano-Contador⁷, Henriette Engen Berg⁸, Frøydis Sved Skottvoll⁸, Francisco Moris⁴, Óscar Zaragoza^{4,9}, Pierre Chymkowitch³, Ignacio Garcia,^{10,#} and Jorrit M. Enserink^{1,2,3,#}

1. Department of Molecular Cell Biology, Institute for Cancer Research, The Norwegian Radium Hospital, Montebello, 0379 Oslo, Norway

2. Centre for Cancer Cell Reprogramming, Institute of Clinical Medicine, Faculty of Medicine, University of Oslo, 0318 Oslo, Norway.

3. Section for Biochemistry and Molecular Biology, Faculty of Mathematics and Natural Sciences, University of Oslo, 0316 Oslo.

4. EntreChem SL, Vivero Ciencias de la Salud, Calle Colegio Santo Domingo Guzmán, s/n 33011 Oviedo (Asturias), Spain.

5. Present address: Fusoni Componentes SL, Polígono de Argame, c/ Mostayal, 33163 Argame, Spain

6. Present address: Laboratorio de Química Sintética Sostenible (QuimSinSos), Departamento de Química Orgánica e Inorgánica, (IUQOEM), Centro de Innovación en Química Avanzada (ORFEO-CINQA), Facultad de Química, Universidad de Oviedo, E33071 Oviedo, Spain.

7. Mycology Reference Laboratory. National Centre for Microbiology. Instituto de Salud Carlos III. Carretera Majadahonda-Pozuelo, Km2. Majadahonda 28220, Madrid, Spain

8. Department of Chemistry, University of Oslo, Blindern, NO-0315 Oslo, Norway

9. Center for Biomedical Research in Network in Infectious Diseases, CB21/13/00105, Instituto de Salud Carlos III, Madrid, Spain.

10. Department of Bacteriology, Norwegian Institute of Public Health, Oslo, Norway

#Correspondence to: ignacio.garciallorente@fhi.no or jorrit.enserink@ibv.uio.no

Abstract

Fungal infections are a growing global health concern due to the limited number of available antifungal therapies as well as the emergence of fungi that are resistant to first-line antibiotics, particularly azoles and echinocandins. Development of novel, selective antifungal therapies is challenging due to similarities between fungal and mammalian cells. An attractive source of potential antifungal treatments is provided by ecological niches co-inhabited by bacteria, fungi and multicellular organisms, where complex relationships between multiple organisms have resulted in the evolution of a wide variety of selective antibiotics. Here, we characterized several analogs of the one such natural compound, Collismycin A. We show that NR-6226C has antifungal activity against several pathogenic *Candida* species, including *C. albicans* and *C. glabrata*, whereas it only has little toxicity against mammalian cells. Mechanistically, NR-6226C selectively chelates iron, which is a limiting factor for pathogenic fungi during infection. As a result, NR-6226C treatment causes severe mitochondrial dysfunction, leading to formation of reactive oxygen species, metabolic reprogramming and a severe reduction in ATP levels. Using an *in vivo* model for fungal infections, we show that NR-6226C significantly increases survival of *Candida*-infected *Galleria mellonella* larvae. Finally, our data indicate that NR-6226C synergizes strongly with fluconazole in killing *C. albicans*. Taken together, NR-6226C is a promising antifungal compound that acts by chelating iron and disrupting mitochondrial functions.

Importance statement

Drug-resistant fungal infections are an emerging global threat, and pan-resistance to current antifungal therapies is an increasing problem. Clearly, there is a need for new antifungal drugs. In this study, we characterized a novel antifungal agent, the Collismycin analog NR-6226C. NR-6226C has a favorable toxicity profile for human cells, which is essential for further clinical development. We unraveled the mechanism of action of NR-6226C and found that it disrupts iron homeostasis and thereby depletes fungal cells of energy. Importantly, NR-6226C strongly potentiates the antifungal activity of fluconazole, thereby providing inroads for combination therapy that may reduce or prevent azole resistance. Thus, NR-6226C is a promising compound for further development into antifungal treatment.

Introduction

Fungal infections range from relatively benign infections of the skin and mucosal tissues to life-threatening invasive infections, affecting more than a billion people worldwide and killing over 1.5 million people annually¹. Infections with *Candida* species, such as *C. albicans* and *C. glabrata*, are among the most common human fungal infections¹. Among the main antifungals that are used in the clinic today are azoles, echinocandins, and polyenes². Azoles block cell membrane synthesis by inhibiting the ergosterol biosynthesis enzyme, lanosterol 14- α -demethylase (encoded by *ERG11* in *Candida*), whereas echinocandins disrupt cell wall synthesis via noncompetitive inhibition of the (1,3)- β -d-glucan synthase enzyme encoded by *FKS* genes. However, long-term use of antifungals has resulted in a steadily increasing prevalence of antifungal resistance, which is a major global

health concern³. Therefore, there exists a need for new antifungal drugs for treatment of antifungal-resistant *Candida* and other pathogenic fungal species.

During evolution, dynamic interactions that occur in ecological niches co-inhabited by bacteria and fungi have resulted in formation of a wide range of antimicrobial compounds that are a rich source of potential antifungal treatments⁴. One example of a complex multi-organism interaction occurs in colonies of leafcutter ants, which form a mutually beneficial symbiotic relationship with a specialized fungus and with various antibiotics-producing bacteria, including *Streptomyces spp.* The ants provide freshly cut leaves to the fungus, resulting in a fungal garden. The fungi produce specialized fungal structures that serve as a key food source for the ant larvae⁵. The bacteria, which grow on the cuticle of the ants, synthesize antifungal compounds such as candicidin or nystatin that protect the fungal garden from invasion by parasitic fungi, which can ruin the fungal garden and destroy the colony^{6,7}. Another such compound is Collismycin A (ColA; Fig. 1A), which is produced by *Streptomyces* and which has antimicrobial activity against several fungi, including *C. albicans*⁸. However, ColA also possesses substantial cytotoxic activities against mammalian cells⁹, rendering it less useful as antifungal therapy.

In this study, we characterized the antifungal activity and overall toxicity of several ColA analogs and identified a compound with antifungal activity against clinical isolates of azole- and echinocandin-resistant *Candida* species, but with reduced toxicity against mammalian cells. Our results suggest that this ColA analog is a promising candidate for development into a treatment for infections caused by wild-type and drug-resistant *Candida* species.

Results

Identification of Collismycin A analogs with increased antifungal activity

To investigate the potential use of novel ColA analogs as selective antifungal agents, we generated a compendium of 26 collismycin-related compounds (Suppl. Fig. S1A; see Methods for details). To determine selectivity against pathogenic fungi, we screened these compounds against a panel of *Candida* strains and two human cell lines. We used *Candida albicans* (CCUG32723), *Candida glabrata* (ATCC15545), as well as an echinocandin-resistant *C. glabrata* strain (*fksl-L662W*; JEY12725) and a fluconazole-resistant *C. glabrata* strain (FL-256; JEY12726), both of which were isolated at Oslo University Hospital (Suppl. Table S1). First, we studied the proliferation of these fungi on increasing concentrations of the starting compound, ColA. ColA reduced the proliferation of all pathogenic yeasts tested (Fig. 1B), with the clinical isolate FL-256 showing the greatest sensitivity with a mean half maximal effective concentration (EC_{50}) of $12.66 \pm 2.55 \mu\text{M}$ (Suppl. Fig. S2 and S3A). We then tested the other analogs and found that the antifungal activity was lost for most of the compounds (Suppl. Fig. S2). However, three compounds, NR-6226C, NR-6226K, and NR-6226V inhibited the growth of *Candida spp* more potently than ColA (Fig. 1C, Suppl. Fig. S3A and B). NR-6226C appeared to be the most efficacious, yielding an EC_{50} at least 12 times greater than ColA-treated strains, whereas treatment with either NR-6226K or NR-6226V resulted in an inhibitory effect that was approximately 6 times greater than ColA (Suppl. Fig. S3B). Importantly, the two clinically isolated antibiotic-resistant *C. glabrata* strains were also sensitive to these compounds (Fig. 1C and Suppl. Fig. S3B). These results encouraged us to further explore the potential of these drugs for further development into antifungal therapy.

For any novel antifungal compound to have a clinical application, it is essential that there exists a therapeutic index, i.e. fungi should be more sensitive to the compound than mammalian cells. Therefore, we tested the sensitivity of HS-5 and HEK-293 fibroblast cell lines to NR-6226C, NR-6226V and NR-6226K. EC_{50} values of NR-6226C were $37.05 \pm 6.94 \mu\text{M}$ and $28.68 \pm 8.32 \mu\text{M}$ in HEK-293 and HS-5 cell lines, respectively (Fig. 1D and Suppl. Fig. S3C), and although we noticed morphological changes at very high concentrations of NR-6226C (100 μM ; Suppl. Fig. S4), these data suggest the existence of a potential therapeutic window. Treatment with NR-6226K

appeared to have no discernible effects on the proliferation of either cell lines; however, closer inspection revealed precipitation of the compound (Suppl. Fig. S4), rendering NR-6226K less useful in physiological settings. We found that NR-6226V strongly attenuated cell proliferation of HS-5 cells ($10.49 \pm 1.42 \mu\text{M}$), and to a somewhat lesser extent also that of HEK-293 cells, limiting its usefulness as a potential antifungal treatment (Fig. 1D and Suppl. Fig. S3C). Together, these results show that the ColA analogs NR-6226C and NR-6226K have antifungal activity exceeding that of ColA, with relatively low toxicity against mammalian cells. For the remainder of the study, we decided to focus on NR-6226C due to its superior activity compared to ColA, its better solubility, its favorable toxicity profile for human cells, and its effectiveness against antifungal-resistant *C. glabrata* strains.

ColA analogue NR-6226C is a Fe^{2+} chelator

ColA has been previously shown to have iron-chelating properties through the formation of a complex containing two ColA molecules bound to a single atom of either Fe^{2+} or Fe^{3+} , and the three nitrogen atoms in ColA have been proposed to facilitate binding of the iron atom to form a tridentate chelator^{9,10,11}.

To test whether iron chelation is important for the inhibitory effect of ColA on fungal growth, we tested the antifungal activity of Collismycin H (ColH), which is a ColA derivative that contains a hydroxymethyl instead of a methyl-imine moiety and therefore cannot chelate $\text{Fe}^{2+/3+}$ (Fig. 2A, *right*). As shown in Figure 2A, ColH lacked antifungal activity even at very high concentrations, showing that the three nitrogen atoms in ColA are essential for antifungal activity. While not conclusive by themselves, these data do support the idea that ColA antifungal activity and its derivatives is mediated through iron chelation. Given that some metal chelators can be promiscuous and bind different metals with different affinities¹², we tested which metals can be bound by NR-6226C using high-performance liquid chromatography and mass spectrometry. These nano liquid chromatography experiments showed that similar to ColA, NR-6226C forms a 2:1

compound-iron complex, where it preferentially binds Fe^{2+} in a concentration-dependent manner, and to a much lesser extent also Fe^{3+} (Fig. 2B and Suppl. Fig. S5).

Next, we determined whether chelation of metal ions is important for antifungal activity of NR-6226C. We performed growth assays of WT *C. glabrata* with NR-6226C in absence or presence of 5 μM of various metal salts. As expected, the addition of FeCl_2 significantly increased the EC_{50} of NR-6226C (Fig. 2C). The same was observed for FeCl_3 , which is likely due to the fact that Fe^{3+} can be converted into Fe^{2+} by ferric reductases. Surprisingly, Cu^{2+} and Zn^{2+} , which hardly bind NR-6226C even at high concentrations, also significantly increased the EC_{50} of NR-6226C. The ameliorating effect of Cu^{2+} and Zn^{2+} on inhibition of fungal growth by NR-6226C may be caused by mismetallation, i.e. inactivation of a protein due to binding of a non-cognate metal^{13,14}, which has previously been shown to induce a compensatory response that promotes uptake of Fe^{2+} ¹⁵. Overall, these findings indicate that NR-6226C inhibits the proliferation of *Candida* strains mainly through iron chelation.

Treatment with NR-6226C induces an iron starvation response

We wished to gain more insight into the physiological response of fungal pathogens to NR-6226C treatment by studying changes in gene expression programs. To avoid potential secondary effects of long-term drug treatment, we treated *C. glabrata* cells for 1 hour with 10 μM NR-6226C and analyzed changes in mRNA levels by RNA sequencing. For comparison, we also included treatment with 5 μM Dp44mT, which is a known iron chelator¹⁶. We found that 224 genes were significantly upregulated and 220 genes were significantly downregulated upon NR-6226C treatment (Suppl. Table S2). Interestingly, genes that were significantly upregulated by NR-6226C included genes involved in the response to iron starvation and the oxidative damage response, such as *TRR1*, *HMX1*, and *HBNI*, which encode thioredoxin, heme oxygenase and an oxidoreductase, respectively (Fig. 3A). Other examples of genes induced by NR-6226C were *CAGL0G06798g*, which is a homolog of *Saccharomyces cerevisiae* (*Sc*) *LSO1* and known to be induced by iron

starvation¹⁷; *CAGL0L11990g*, a homolog of *Sc GRX4*, encoding an iron-response regulating enzyme with glutathione-dependent oxidoreductase and glutathione S-transferase activities¹⁸; and *CAGL0D04708g*, a major copper transporter that is transcriptionally induced at low copper levels¹⁹. Genes that were downregulated by NR-6226C include genes encoding proteins with iron-sulfur clusters, such as *CAGL0M00374g*, which encodes H₂S-NADP oxidoreductase; the succinate dehydrogenase-encoding gene *SDH2*, and the aconitase-encoding gene *ACO1*, which are preferentially expressed in the presence of sufficiently high iron levels²⁰. Other downregulated genes are *DUG3*, which mediates degradation of glutathione²¹; and the detoxifying metallothionein gene *MT-I*, which is induced by high levels of metal ions²². As expected, gene ontology (GO) analysis revealed that these downregulated genes function in mitochondrial and enzymatic functionalities, such as lyase activity, oxidoreductase activity, and catalytic activity (Fig. 3C). Together, these findings strongly suggest that NR-6226C treatment induces an iron starvation response and possibly also an oxidative damage response.

To further characterize the transcriptional response to NR-6226C, we compared it to that of the known metal chelator Dp44mT, which chelates both copper and iron^{16,23}. As expected, there was a substantial correlation between the transcriptional responses elicited by these two compounds, showing an overlap of 58% and 55% of genes that were significantly upregulated and downregulated, respectively (Supp. Fig. S6A-C). Genes encoding iron-binding metalloenzymes or enzymes containing iron-sulfur clusters were strongly downregulated by NR-6226C and Dp44mT treatment, such as *SDH2*, *ACO1/2*, *CAGL0E05676g* (*Sc TYW1*), *LYS9*, as well as the iron-sulfur assembly coding gene *CAGL0G03905g* (*Sc ISA1*). These results further support the idea that NR-6226C is a potent iron chelator.

The iron starvation response in *S. cerevisiae* has been reported to involve the transcription factors Aft1 and Aft2^{24,25,26}. Given that *C. glabrata* is very closely related to *S. cerevisiae*²⁷, and because it is challenging to create gene deletions in *C. glabrata*, we used *S. cerevisiae* to test

whether some of the responses to NR-6226C are mediated by Aft1. For comparison, we included Dp44mT as a control. Interestingly, treatment of wild-type (WT) *Sc* cells with NR-6226C resulted in robust activation of Aft1/2 target genes *FTR1*, *FET3*, *HMX1*, and *LSO1*, while there was no effect of NR-6226C on expression of control 5S RNA (Fig. 3D). Importantly, deletion of the *AFT1* gene attenuated the transcriptional responses upon treatment with NR-6226C (Fig. 3D).

Taken together, we conclude that treatment with NR-6226C induces a strong iron starvation response in *Candida sp.* and *S. cerevisiae*. Given that iron is a limiting factor in microbial pathogenesis²⁸, NR-6226C is a promising candidate for treatment of microbial infections.

Treatment with NR-6226C induces ROS formation

Low levels of iron can result in dysregulation of iron-dependent processes, including oxidative phosphorylation, thereby leading to mitochondrial dysfunction and generation of reactive oxygen species (ROS). Given our finding that NR-6226C treatment results in activation of genes involved in oxidative damage response, we tested whether NR-6226C induces formation of ROS in *C. glabrata*. Fungal cells were preloaded with the fluorescent ROS sensor H₂DCF-DA, after which we measured changes in fluorescence after 3, 6, and 24 hours after exposure to either DMSO, 30 μ M NR-6226C, or 75 μ M H₂O₂. As shown in Figure 4A, after 24h of treatment with NR-6226C there was a significant increase in ROS production compared to cells treated with DMSO, demonstrating that NR-6226C treatment induces ROS production.

We hypothesized that NR-6226C-induced ROS formation was responsible for the antifungal effect of NR-6226C. However, treatment with the ROS scavenger N-acetyl-cysteine (NAC) did not rescue the negative effect of NR-6226C on fungal cell viability (Suppl. Fig. S7B), suggesting that ROS production is not the primary cause of death. Rather, inactivation of crucial iron-dependent metabolic processes may underlie the effect of NR-6226C, such as loss of ATP production and reduced synthesis of various essential metabolites. To test whether NR-6226C affects metabolism and cellular ATP production, we treated *C. glabrata* with increasing concentrations of NR-6226C.

Equal numbers of cells were harvested for each concentration point and relative ATP levels were quantified using Cell-Titer Glo²⁹. As shown in Figure 4B, treatment with NR-6226C resulted in a concentration-dependent reduction in luminescence, reflecting decreased metabolic activity and loss of cellular ATP production. This suggests that treatment with NR-6226C has a negative effect on mitochondrial activity. We therefore labeled *C. glabrata* cells with the fluorescent dye MitoTracker Red CMXRos, which fluoresces preferentially upon entry into metabolically active mitochondria. Interestingly, after 24h treatment with NR-6226C, cells clearly exhibited lower fluorescence intensities compared to cells treated with DMSO (Fig. 4C), confirming that NR-6226C impairs mitochondrial activity. Taken together, NR-6226C causes significant mitochondrial dysfunction in *C. glabrata* cells, leading to ROS formation, loss of metabolic activity, and depletion of cellular ATP levels.

26C has antifungal activity in an in vivo infection model

To investigate whether NR-6226C has antifungal activity *in vivo*, we employed *Galleria mellonella* as an infection model. Recently, *G. mellonella* has emerged as a robust infection model for studying virulence and antimicrobial therapies against infectious agents³⁰. *Galleria* is capable of mounting an efficient innate immune response against pathogenic fungi that is broadly comparable with the mammalian antifungal immune response³⁰. In addition to obvious ethical issues, another important advantage of *G. mellonella* over mice is that there are no concerns of introducing pathogens into breeding facilities, which is a major challenge with mouse models.

In brief, *G. mellonella* larvae were infected with WT *C. glabrata*, followed by incubation in the presence of either DMSO (untreated control) or 30 μ M NR-6226C. As shown in Figure 5A and B, treatment with NR-6226C significantly improved the survival of *G. mellonella* larvae infected with either wild-type or azole-resistant *C. glabrata*. We conclude that NR-6226C is a selective antifungal agent with *in vivo* activity that bypasses azole resistance.

Synergistic antifungal effect of NR-6226C and fluconazole

One approach to countering antifungal resistance that has recently gained interest is combination therapy³¹, where the simultaneous use of two drugs produces either an additive or synergistic effect to impede cellular growth. This strategy could prove beneficial in many circumstances. For example, drugs may be used at lower concentrations to inhibit fungal growth, thereby preventing general toxicity to the host. Combinations of drugs that target different fungal processes may also reduce the likelihood of development of acquired resistance to antifungal drugs. We therefore tested NR-6226C in combination with fluconazole, a first-line antifungal that is often clinically ineffective against common nosocomial infections, such as certain *Candida spp.* Interestingly, while wild-type *C. albicans* was only susceptible to high concentrations of fluconazole, combination of fluconazole with NR-6226C resulted in a significantly stronger impairment of fungal proliferation than either drug alone (Fig 6A). To further determine whether the combinatorial effects of these compounds were synergistic, we analyzed our data using the SynergyFinder Plus package³² for R using the Bliss model for synergistic interactions³³. This revealed strong synergy between NR-6226C and fluconazole (Fig. 6B and Supp. Fig. S8A). Although *C. glabrata* is more sensitive to NR-6226C than *C. albicans*, we did not observe substantial additive or synergistic effects of drug combinations (Suppl. Fig. S8B-G). This might be expected, because *C. glabrata* exhibits intrinsically low susceptibility to fluconazole³⁴. While analysis of combinations between NR-6226C and a broader panel of antifungal drugs against a spectrum of pathogenic fungi will be the focus of future studies, these results demonstrate that NR-6226C strongly potentiates the antifungal activity of fluconazole against *C. albicans*.

Discussion

A key challenge in combating fungal resistance is the limited selection of antifungal drugs that are clinically available. Our study revealed that the compound NR-6226C has potential as an

antifungal agent. Pathogenic fungi require iron for infection, but the concentration of free iron in mammals is very low ($\sim 10^{-9}$ M) due to sequestration by iron-binding and iron-containing compounds³⁵. In our study, we found that NR-6226C prevented *Candida* proliferation *in vitro* and *in vivo* by chelating iron. This resulted in activation of iron starvation response genes such as *FTR1*, *HMX1*, *LSO1* and *FET3* in a Aft1/2-dependent manner. *FET3* encodes a high-affinity iron importer, which mediates iron uptake from the extracellular environment²⁵ and it has been proposed that *LSO1* encodes a cytoplasmic protein involved in intracellular iron transport¹⁷. Furthermore, to overcome the limiting iron availability in their host, pathogenic fungi increase the Aft1-dependent expression of the heme oxygenase Hmx1, which facilitates degradation of heme from the host, thereby releasing iron for subsequent metabolic use^{36,37,38}. Our results indicate that iron chelation by NR-6226C has a direct effect on fungal iron supply, thereby affecting iron metabolism and subsequent intracellular regulation.

Other than the activation of genes regulated by the Aft1 iron-sensory system, our RNA sequencing data revealed the downregulation of enzymatic activities and other genes linked to mitochondrial functions. These findings are consistent with previous yeast studies showing that iron homeostasis is tightly coordinated with metabolic processes³⁹. Given that multiple iron-dependent metabolic processes occur in mitochondria, cells likely redirect the limited amount of available iron to essential iron-dependent pathways to maintain homeostasis. The significant decrease of the expression of Fe-S cluster-dependent genes encoding *H₂S-NADP* oxidoreductase, *Sdh2*, and *Aco1* (Fig. 3A), which have known functions in the mitochondrial respiratory chain^{40,41}, supports this hypothesis. Further evidence that NR-6226C perturbs iron homeostasis is provided by induction of *CAGL0L11990g*, a homolog of *Sc GRX4*. Grx4 is a cytosolic monothiol glutaredoxin that not only functions in concert with Fe-S clusters as an iron sensor, but which is also involved in Fe-S cluster biogenesis in the mitochondria^{42,43}. Although we have not studied the molecular mechanism by which iron depletion results in downregulation of these genes in *C. albicans*, studies in *S. cerevisiae* have revealed that this is at least in part due to Cth2-dependent post-transcriptional degradation and

translational-repression of these mRNAs^{44,45}. Given the high degree of conservation between these *Saccharomyces* and *Candida*, it is likely that similar mechanisms operates in *C. albicans*.

While NR-6226C has a strong preference for Fe²⁺, the addition of Zn²⁺ and Cu²⁺ to the extracellular medium also countered its antifungal activity, even though these metal ions do not efficiently interact with NR-6226 *in vitro* even at high concentrations. Although we cannot exclude the possibility that the antifungal effect of NR-6226C is mediated by sequestration of trace amounts of Zn²⁺ and Cu²⁺ *in vivo*, we believe it is more likely that these metals induce a stress response that compensates for iron chelation by NR-6226C. Indeed, high concentrations of heavy metals cause mismetallation^{14,46}, which can induce an iron starvation response that results in increased uptake of iron from the environment by microorganisms¹⁵.

It is well known that persistent iron deprivation leads to increased levels of oxidative stress over time⁴⁷. Similarly, exposure to NR-6226C resulted in a significant increase in ROS production. However, ROS production did not appear to be the direct cause for loss of cell proliferation, as treatment with ROS scavengers did not rescue the effects of NR-6226C. Although long-term NR-6226C-induced ROS production may have a deleterious effect on fungal cells, the immediate effects of NR-6226C may rather be due to impaired metabolic activity and ATP production. These findings mirror previous studies that linked iron metabolism and mitochondrial activity to fungal pathogenicity in *C. albicans*, *C. auris*, and *Aspergillus fumigatus*^{48,49,50,51}. These links between mitochondria and virulence include lipid biosynthesis, mitochondrial fusion, mitochondrial respiration, and calcium signaling, and indicate that therapies affecting mitochondrial function could prove to be an effective form of antifungal therapy. Indeed, a recent study demonstrated that iron homeostasis and mitochondrial activities may be important targets for treatment of *Candida auris*⁵⁰.

An interesting finding of our study is that NR-6226C strongly synergizes with fluconazole in blocking proliferation of *C. albicans*. Drug combinations can have several advantages, such as the need for lower concentrations of drugs to achieve the desired effect, thereby reducing the risk of off-target toxicity towards host cells. Another advantage is that it is likely more difficult for cells to develop simultaneous resistance against two drugs with separate mechanisms of action. Although more comprehensive drug combination testing against a range of pathogenic fungi will be the focus of future studies, it will be particularly interesting to test combinations of NR-6226C with azoles, because these antifungals exert their effect by blocking ergosterol biosynthesis, and expression of genes required for ergosterol synthesis is strongly dependent on iron availability⁵².

In conclusion, NR-6226C is an interesting lead compound for further development into antifungal therapy. In future studies, it will be of interest to study the effects of NR-6226C on other fungal pathogens such as *Aspergillus fumigatus* and *Cryptococcus neoformans*, and studies in mammalian models will be important to further characterize the *in vivo* potential of NR-6226C, and further studies may be needed to optimize its antifungal activity and ADME profile.

Methods

Culture Maintenance

Candida spp. and *S. cerevisiae* strains used in this study are listed in Suppl. Table S1. Cells were stored at -80°C in YPD (1% Yeast extract, 2% Peptone, and 2% Dextrose) media containing 20% glycerol. For experimental use, cells were streaked out on solid YPD plates (with 2% agar) and incubated at 30°C for 2-3 days until single colonies appeared. The solid plates were stored at 4°C for up to three weeks until new cells were streaked out again.

For experiments, yeast cells were routinely cultured in YPD media and incubated at 30 °C overnight. The cultures were refreshed the next day for 2-3 hours in YNB-iron (0.69% Yeast Nitrogen Base without Amino Acids or Iron, and 2% Dextrose) media, unless stated, at the same temperature. After incubation, the optical density (OD₆₀₀) of the cell cultures were measured and adjusted to 0.15.

Compound Screening

The Collismycin compounds were provided by EntreChem (Spain) at stock solutions of 20 mM in dimethyl sulfoxide (DMSO) and stored at -20°C. For serial dilutions of the compounds in clear 96-well plates, intermediate stock concentrations of 100 µM of compound (100 µl for each technical replicate) were prepared in YNB-iron media unless otherwise specified and 1:1 serial dilutions were performed using a multi-channel pipette. 50 µl of cell culture (OD₆₀₀ = 0.15) were added to the wells containing the compounds, for a total volume of 150 µl. The final concentrations in the wells (with three technical replicates) were equivalent to 0.5 µM, 1 µM, 2.1 µM, 4.2 µM, 8.4 µM, 16.7 µM, 33.3 µM, and 66.6 µM. The OD₆₀₀ of each well was quantified and the plates were incubated inside plastic bags (to prevent evaporation) at 30°C for 24 hours, after which the final OD₆₀₀ was measured.

Human Cell Lines

HEK-293 and HS-5 cells were stored at -180 °C, in freezing medium (50% DMEM, 40% Fetal bovine serum and 10% DMSO) for long-term storage. For experimental use, the cells were thawed and then cultured using Dulbecco's Modified Eagle's Medium (DMEM) containing 10% fetal serum albumin and 1% Penicillin/Streptomycin.

Cell-Titer Glo Assay

1:1 serial dilutions of the compounds were performed in 96-well plates to obtain a final concentration range (three technical replicates each) equivalent to 0.781 μM , 1.562 μM , 3.125 μM , 6.25 μM , 12.5 μM , 25 μM , 50 μM and 100 μM . Each well had a total volume of 100 μl , consisting of 50 μl compound and 50 μl seeded cells (10,000 cells per well). The plates were placed inside plastic bags to prevent evaporation and incubated at 37°C, 5% CO_2 for 24 hours. 100 μl of Cell Titer-Glo 2.0 (Promega, G9243) was added to each well and then measured for luminescence using a Synergy 2 Gen 5 plate reader from BioTek.

Nano liquid chromatography-mass spectrometry

Liquid chromatography-mass spectrometry (LC-MS) grade water, acetonitrile, and formic acid ($\geq 99\%$) were purchased from VWR (Radnor, PA, U.S). NR-6226C (Terpyridine (2,2':6',2''-Terpyridine, 98%)) was purchased from Sigma Aldrich (now Merck, KGaA, Darmstadt Germany). Metal solutions containing 2 μM , 4 μM , 6 μM and 10 μM CaCl_2 , CuCl_2 , FeCl_2 , FeCl_3 , MgCl_2 , MnCl_2 , and ZnCl_2 in 0.1 % formic acid in LC-MS grade water (hereby referred to as 0.1 % formic acid) was made. A stock solution of terpyridine (100 μM) in 0.1% formic acid was made and added to the metal solutions to a final concentration of 10 μM terpyridine. All metal solutions in addition to a blank sample containing 10 μM terpyridine in 0.1% formic acid were subjected to nano liquid chromatography (nanoLC-MS) analysis. NanoLC-MS was performed using an nLC EASY 1000 pump connected to a Q-Exactive mass spectrometer (MS) with a Nanospray FlexTM ion source (all from Thermo Fisher Scientific, Waltham, MA, U.S). For separation, a 50 μm (inner diameter) x 5 cm in-house packed nanoLC column containing 2.6 μm Accucore C18 particles (80 Å) was used. The column was packed according to the protocol from Berg, et al⁵⁶. Trapping of analytes prior to separation was performed on-line using a 2 cm Acclaim PepMap with 3 μm particles (100 Å). All columns and packing materials were obtained from Thermo Fisher Scientific. The mobile phases consisted of 0.1% formic acid (A) and 90/10/0.1 acetonitrile/water/FA (v/v/v) (B). A linear gradient from 5-20% B in 10 min was employed, and trapping was performed by 100% mobile phase A.

Equilibration was performed using 2 μ l and 3 μ l 100% A for the trapping column and analytical column, respectively. The flow rate was set to 250 nl/min and the injection volume used was 1 μ l.

The MS was operated in positive mode using single ion monitoring (SIM). A mass-to-charge ratio (m/z) of 234.1 (M+H) was used for terpyridine and an m/z of 261.1 (2M+Fe)²⁺ was used for terpyridine bound to Fe²⁺. All data processing was performed by using the XCaliburTM Software (Thermo Fisher Scientific).

Cell proliferation with metal supplementation

NR-6226C (Sigma-Aldrich) was prepared at a stock solution of 20 mM in DMSO. 1 mM stocks of metal solutions (CaCl₂, CuCl₂, FeCl₂, FeCl₃, MgCl₂, MnCl₂, and ZnCl₂) were prepared in mQH₂O and autoclaved. Serial dilutions of NR-6226C using YNB-iron media, were performed in clear 96-well plates to obtain final concentrations of 0.78 μ M, 1.56 μ M, 3.13 μ M, 6.25 μ M, 12.5 μ M, 25 μ M, 50 μ M and 100 μ M. 50 μ l of either mQH₂O or metal solution were added to each well, including 50 μ l of refreshed *C. glabrata* WT with an adjusted OD₆₀₀ of 0.15. The 96-well plates were measured at OD₆₀₀ and incubated in plastic bags to prevent evaporation for 24 hours, after which the final OD₆₀₀ was measured.

RNA Sequencing

Cells from overnight precultures in YPD medium were reinoculated in fresh YNB medium without iron and grown until OD₆₀₀ = 0.4. Cultures were then treated for 1 hr with DMSO (vehicle) or 10 μ M NR-6226C or 5 μ M Dp44mT. Cells were collected from three independent experiments, snap frozen and stored at -80°C. Total RNA purification was performed as previously described using the RNeasy Mini Kit (74104, Qiagen) and gDNA removal was made using the RNase-Free DNase Set kit (79254, QIAGEN). RNA quantification and quality control were done with a Tape Station 4150 (Agilent)⁵⁷.

Library preparation and sequencing were performed at GenomEast (IGBMC, Illkirch, France). One DMSO 0.05% sample was excluded from the analysis due to insufficient quality. The library was sequenced on Illumina HiSeq 4000 sequencer as Single-Read 50 base reads following Illumina's instructions. Reads were pre-processed in order to remove adapter, polyA and low-quality sequences (Phred quality score below 20). Reads shorter than 40 bases were discarded from further analysis. These pre-processing steps were performed using cutadapt version 1.10⁵⁸. Image analysis and base calling were performed using RTA 2.7.7 and bcl2fastq2.17.1.14. Adapter dimer reads were removed using DimerRemover (<https://sourceforge.net/projects/dimerremover/>). The quality of the RNAseq reads was examined using FastQC 0.11.2 (<http://www.bioinformatics.babraham.ac.uk/projects/fastqc/>) and FastQScreen 0.5.1 ([http://www.bioinformatics.babraham.ac.uk/projects/fastq screen/](http://www.bioinformatics.babraham.ac.uk/projects/fastq%20screen/)). Reads were mapped onto the ASM254v2 assembly of *Candida glabrata* genome using STAR version 2.5.3a⁵⁹. Gene expression quantification was performed from uniquely aligned reads using htseq-count version 0.6.1p1⁶⁰, with annotations from Ensembl fungi version 50 and union mode. Only non-ambiguously assigned reads have been retained for further analyses. Read counts were normalized across samples with the median-of-ratios method⁶¹. Comparisons were implemented in the Bioconductor package DESeq2 version 1.16.1 using the test for differential expression⁶². Genes with no p-value corresponded to genes with high Cook's distance that were filtered out. P-values were adjusted for multiple testing using the Benjamini and Hochberg method⁶³. Genes with no adjusted p-value correspond to genes filtered out in the independent filtering step in order to remove reads from genes that have no or little chance of showing significant evidence of differential expression.

RT-qPCR

WT and *atf1Δ* cells were grown overnight in CSM. Cells were then diluted 10 times and grown until log phase at which point 10 μM of NR6226C or Dp44mt were added to the medium, either in absence or presence of 5 μM of FeCl₂ as indicated. After 24 hrs of incubation cells were collected

and total RNA purification, reverse transcription and RT-qPCR were performed as previously described with minor modifications^{57,64}. Briefly, total RNA was purified using the RNeasy Mini Kit (74104, Qiagen) and reverse transcription was performed using the QuantiTect Reverse Transcription Kit (205311, Qiagen). RT-qPCR experiments were done using the HOT FIREPol® EvaGreen® qPCR Mix (08-36-00001-10, Solis Biodyne) and a LightCycler® 96 System (Roche). Expression of the 5S gene was used as a control. Primers used in RT-qPCR experiments:

FTR1: GATTGGGTTCTTGAGTAGAAG and GAGCCCTGTGTGGTAATA

FET3: GTCAATATGAAGACGGGATG and CCACTCACTAAGCGATAAAG

HMX1: CCAGAGATGCCACAATA and CATAATAGTACGCCAGAATACC

LSO1: AGAAGAAGCTGATTATGGAAC and CTGCCTCCCTTACCTAAA

5S: GTTGCGGCCATATCTACCAGAAAG and CGTATGGTCACCCACTACTACT

Reactive oxygen species (ROS) assay

Our method was adapted using the protocol developed by James, et al.⁶⁵ for ROS assessment in yeast. The overnight culture was refreshed in YNB-iron media for 2-3 hours and then adjusted to $OD_{600} = 0.5$ in 100 ml YNB-iron. The cells were washed by centrifugation for 5 minutes at 4300 rpm, followed by aspiration of the supernatant and a re-suspension of the cell pellets in 20 ml phosphate buffer solution (PB, 0.1M, pH 7.4). 1 ml of cell suspension was aliquoted to Falcon tubes and then centrifuged again to remove the supernatant. The general oxidative stress indicator, CM-H₂DCFDA (Thermo Fisher), was prepared in a stock solution of 1 mM DMSO and then diluted to 10 μ M using PB solution. The cell pellets were pre-loaded with the dye by re-suspension in 10 μ M CM-H₂DCFDA and then incubated in the dark at 30°C for 30 minutes. Cell pellets that were used for a negative control of the dye were resuspended in PB. After incubation, the samples were centrifuged at 4300 rpm for 2 minutes and followed by aspiration of the supernatant. The cell pellets were then treated with either DMSO, 30 μ M NR-6226C, or 75 mM hydrogen peroxide (H₂O₂), including the negative controls. The samples were placed in a 30°C incubator and then

measured for OD₆₀₀ after 3, 6 and 24 hours of incubation, after which an equal number of cells (OD₆₀₀ = 0.5, 1 ml) were harvested. The harvested cells were centrifuged at 6000g for 10 minutes, with the supernatant aspirated and the cells resuspended in PB solution. The fluorescence of the samples was measured in a black, clear-bottom 96-well plate using a spectrofluorometer.

Cell-Titer Glo assay in *Candida*

A volume of 500 ml cell culture in YNB-iron (OD₆₀₀ = 0.15) was mixed to a final concentration of 1 ml DMSO, 0.625 μM NR-6226C, 2.5 μM NR-6226C, or 10 μM NR-6226C. The samples were incubated for 24 hours at 30°C and then measured for OD₆₀₀. Equal number of cells were harvested in each experiment according to an OD₆₀₀ = 0.1-0.4, then centrifuged at 7000 rpm for 5 minutes to remove the supernatant. The cell pellets were resuspended in 100 μl and then transferred to a 96-well plate. Finally, the samples were treated with 100 μl Cell-Titer Glo and then measured for luminescence using a plate reader.

Mitochondrial staining with MitoTracker

MitoTracker Red CMXRos (Invitrogen) was prepared to a stock solution of 1 mM using DMSO and stored at -20°C. 5 ml *C. glabrata* WT cells of OD₆₀₀ = 0.15 were centrifuged at 7000 rpm for 5 minutes to obtain cell pellets. The cell pellets were resuspended in 5 ml DMSO, 2.5 μM or 10 μM NR-6226C and then incubated at 30°C for 24 hours. The samples were measured for OD₆₀₀, followed by harvesting of cells to an OD₆₀₀ = 0.3 and then centrifuged at 7000 rpm for 5 minutes. The cell pellets were resuspended in 100 nM MitoTracker and incubated at 37°C for 25 minutes. The samples were centrifuged at 7000 rpm for 5 minutes and the resulting pellets were washed with 1 ml milliQ H₂O. The cell pellets were resuspended in milliQ H₂O and then imaged using a Zeiss LSM 710 microscope.

Survival assays in infected *Galleria mellonella* models

Galleria mellonella was obtained from R.J. Mous Livebait (The Netherlands). The larvae were selected by weight (0.2-0.3 grams) and by the absence of dark spots on the cuticle. The larvae were maintained at room temperature, and the day before the experiment they were transferred to the temperature at which the experiment was going to be performed (30 °C). The number of dead caterpillars was scored every day. A group of 20 larvae were incubated with PBS, DMSO and with NR-6226C alone as controls in every experiment. *G. mellonella* were infected with either *C. glabrata* 2001HT or FL-256; *C. glabrata* cells were grown in liquid media (Oxoid) overnight at 30 °C with moderate shaking (150 rpm). Groups of 20 larvae per strain were infected in each experiment. The pro-leg area was cleaned with 70% ethanol using a swab. The larvae were inoculated with 10 µl of *C. glabrata* suspension at 2.5×10^6 cells/ml in PBS containing 50 µg/ml of ampicillin by injection, using a Hamilton syringe with 26-gauge needle in the last left proleg. Within 2 hours of infection, 10 µl of compound solution NR-6226C (15 and 30 µM) was injected into a different pro-leg using the same technique.

Combination treatment with Fluconazole and NR-6226C

Stock solutions of fluconazole (TargetMol) and NR-6226C (20 mM in DMSO) were prepared. Fluconazole (1.56, 3.12, 6.25, 12.5, 25, 50, and 100 µM) and NR-6226C (1, 2, 3, 4, 5, and 10 µM) were dispensed into clear 96-well plates using CERTUS FLEX liquid dispenser. 50 µl of refreshed *Candida* cells in YNB-iron equivalent to $OD_{600} = 0.15$ were added to the wells and measured for OD_{600} . The plates were placed in plastic bags and incubated at 30 °C for 24 hours until the final OD_{600} was measured again.

Data Availability

Data description: RNAseq dataset. Name of the repository: ENA database. Accession number: PRJEB63373.

Acknowledgments

This work was supported by grants from the Norwegian Cancer Society (project numbers 182524 and 208012), the Norwegian Health Authority South-East (2017064, 2017072, 2018012, 2019096), the Research Council of Norway (261936, 301268, 262652). OZ is supported by grant PID2020-114546RB by MCIN/AEI/10.13039/501100011033. Sequencing was performed by the GenomEast platform, a member of the ‘France Genomique’ consortium [ANR-10-INBS-0009]. We thank Cecilie Torp Andersen (Oslo University Hospital) for providing patient-derived *Candida* strains that were used in this work. We also thank all members of the Enserink group and the Knævelsrud group for helpful discussions and feedback.

References

1. Bongomin F, Gago S, Oladele RO, Denning DW. 2017. Global and Multi-National Prevalence of Fungal Diseases-Estimate Precision. *J Fungi (Basel)* 3:57.
2. Robbins N, Caplan T, Cowen LE. 2017. Molecular Evolution of Antifungal Drug Resistance. *Annu Rev Microbiol* 71:753–775.
3. Fisher MC, Alastruey-Izquierdo A, Berman J, Bicanic T, Bignell EM, Bowyer P, Bromley M, Brüggemann R, Garber G, Cornely OA, Gurr SJ, Harrison TS, Kuijper E, Rhodes J, Sheppard DC, Warris A, White PL, Xu J, Zwaan B, Verweij PE. 2022. Tackling the emerging threat of antifungal resistance to human health. *Nat Rev Microbiol* 20:557–571.
4. Vij R, Hube B, Brunke S. 2021. Uncharted territories in the discovery of antifungal and antivirulence natural products from bacteria. *Comput Struct Biotechnol J* 19:1244–1252.
5. De Fine Licht HH, Boomsma JJ, Tunlid A. 2014. Symbiotic adaptations in the fungal cultivar of leaf-cutting ants. *Nat Commun* 5:5675.

6. Haeder S, Wirth R, Herz H, Spiteller D. 2009. Candicidin-producing *Streptomyces* support leaf-cutting ants to protect their fungus garden against the pathogenic fungus *Escovopsis*. *Proc Natl Acad Sci USA* 106:4742–4746.
7. Seipke RF, Grüşchow S, Goss RJM, Hutchings MI. 2012. Isolating Antifungals from Fungus-Growing Ant Symbionts Using a Genome-Guided Chemistry Approach, p. 47–70. *In* *Methods in Enzymology*. Elsevier.
8. Gomi S, Amano S, Sato E, Miyadoh S, Kodama Y. 1994. Novel antibiotics SF2738A,B and C, and their analogs produced by *Streptomyces* sp. *J Antibiot* 47:1385–1394.
9. Garcia I, Vior NM, González-Sabín J, Braña AF, Rohr J, Moris F, Méndez C, Salas JA. 2013. Engineering the Biosynthesis of the Polyketide-Nonribosomal Peptide Collismycin A for Generation of Analogs with Neuroprotective Activity. *Chemistry & Biology* 20:1022–1032.
10. Antonini I, Claudi F, Cristalli G, Franchetti P, Grifantini M, Martelli S. 1981. N*-N*-S* Tridentate ligand system as potential antitumor agents. *J Med Chem* 24:1181–1184.
11. Kawatani M, Muroi M, Wada A, Inoue G, Futamura Y, Aono H, Shimizu K, Shimizu T, Igarashi Y, Takahashi-Ando N, Osada H. 2016. Proteomic profiling reveals that collismycin A is an iron chelator. *Sci Rep* 6:38385.
12. Chohan ZH, Hanif M. 2010. Design, synthesis, and biological properties of triazole derived compounds and their transition metal complexes. *J Enzyme Inhib Med Chem* 25:737–749.
13. Chandrangsu P, Rensing C, Helmann JD. 2017. Metal homeostasis and resistance in bacteria. *Nat Rev Microbiol* 15:338–350.
14. Imlay JA. 2014. The mismetallation of enzymes during oxidative stress. *J Biol Chem* 289:28121–28128.
15. Goff JL, Chen Y, Thorgersen MP, Hoang LT, Poole FL, Szink EG, Siuzdak G, Petzold CJ, Adams MWW. 2023. Mixed heavy metal stress induces global iron starvation response. *ISME J* 17:382–392.

16. Yuan J, Lovejoy DB, Richardson DR. 2004. Novel di-2-pyridyl-derived iron chelators with marked and selective antitumor activity: in vitro and in vivo assessment. *Blood* 104:1450–1458.
17. An X, Zhang C, Sclafani RA, Seligman P, Huang M. 2015. The late-annotated small ORF LSO1 is a target gene of the iron regulon of *Saccharomyces cerevisiae*. *Microbiologyopen* 4:941–951.
18. Pujol-Carrion N, Belli G, Herrero E, Nogues A, de la Torre-Ruiz MA. 2006. Glutaredoxins Grx3 and Grx4 regulate nuclear localisation of Aft1 and the oxidative stress response in *Saccharomyces cerevisiae*. *J Cell Sci* 119:4554–4564.
19. Dancis A, Yuan DS, Haile D, Askwith C, Eide D, Moehle C, Kaplan J, Klausner RD. 1994. Molecular characterization of a copper transport protein in *S. cerevisiae*: an unexpected role for copper in iron transport. *Cell* 76:393–402.
20. Lan C-Y, Rodarte G, Murillo LA, Jones T, Davis RW, Dungan J, Newport G, Agabian N. 2004. Regulatory networks affected by iron availability in *Candida albicans*. *Mol Microbiol* 53:1451–1469.
21. Ganguli D, Kumar C, Bachhawat AK. 2007. The alternative pathway of glutathione degradation is mediated by a novel protein complex involving three new genes in *Saccharomyces cerevisiae*. *Genetics* 175:1137–1151.
22. Mehra RK, Garey JR, Butt TR, Gray WR, Winge DR. 1989. *Candida glabrata* metallothioneins. Cloning and sequence of the genes and characterization of proteins. *J Biol Chem* 264:19747–19753.
23. Lovejoy DB, Jansson PJ, Brunk UT, Wong J, Ponka P, Richardson DR. 2011. Antitumor activity of metal-chelating compound Dp44mT is mediated by formation of a redox-active copper complex that accumulates in lysosomes. *Cancer Res* 71:5871–5880.
24. Philpott CC, Protchenko O. 2008. Response to Iron Deprivation in *Saccharomyces cerevisiae*. *Eukaryot Cell* 7:20–27.

25. Ramos-Alonso L, Romero AM, Martínez-Pastor MT, Puig S. 2020. Iron Regulatory Mechanisms in *Saccharomyces cerevisiae*. *Front Microbiol* 11:582830.
26. Yamaguchi-Iwai Y, Dancis A, Klausner RD. 1995. AFT1: a mediator of iron regulated transcriptional control in *Saccharomyces cerevisiae*. *The EMBO Journal* 14:1231–1239.
27. Marcet-Houben M, Gabaldón T. 2009. The tree versus the forest: the fungal tree of life and the topological diversity within the yeast phylome. *PLoS One* 4:e4357.
28. Cassat JE, Skaar EP. 2013. Iron in infection and immunity. *Cell Host Microbe* 13:509–519.
29. Crouch SP, Kozlowski R, Slater KJ, Fletcher J. 1993. The use of ATP bioluminescence as a measure of cell proliferation and cytotoxicity. *J Immunol Methods* 160:81–88.
30. Lionakis MS. 2011. *Drosophila* and *Galleria* insect model hosts: new tools for the study of fungal virulence, pharmacology and immunology. *Virulence* 2:521–527.
31. Fioriti S, Brescini L, Pallotta F, Canovari B, Morroni G, Barchiesi F. 2022. Antifungal Combinations against *Candida* Species: From Bench to Bedside. *J Fungi (Basel)* 8:1077.
32. Zheng S, Wang W, Aldahdooh J, Malyutina A, Shadbahr T, Tanoli Z, Pessia A, Tang J. 2022. SynergyFinder Plus: Toward Better Interpretation and Annotation of Drug Combination Screening Datasets. *Genomics, Proteomics & Bioinformatics* 20:587–596.
33. Ianevski A, Giri AK, Aittokallio T. 2022. SynergyFinder 3.0: an interactive analysis and consensus interpretation of multi-drug synergies across multiple samples. *Nucleic Acids Research* 50:W739–W743.
34. Oxman DA, Chow JK, Frendl G, Hadley S, Hershkovitz S, Ireland P, McDermott LA, Tsai K, Marty FM, Kontoyiannis DP, Golan Y. 2010. *Candidaemia* associated with decreased in vitro fluconazole susceptibility: is *Candida* speciation predictive of the susceptibility pattern? *J Antimicrob Chemother* 65:1460–1465.
35. Ratledge C. 2007. Iron Metabolism and Infection. *Food Nutr Bull* 28:S515–S523.

36. Kim D, Yukl ET, Moënne-Loccoz P, Ortiz de Montellano PR. 2006. Fungal Heme Oxygenases: Functional Expression and Characterization of Hmx1 from *Saccharomyces cerevisiae* and CaHmx1 from *Candida albicans*. *Biochemistry* 45:14772–14780.
37. Devaux F, Thiébaud A. 2019. The regulation of iron homeostasis in the fungal human pathogen *Candida glabrata*. *Microbiology* 165:1041–1060.
38. Navarathna DHMLP, Roberts DD. 2010. *Candida albicans* heme oxygenase and its product CO contribute to pathogenesis of candidemia and alter systemic chemokine and cytokine expression. *Free Radical Biology and Medicine* 49:1561–1573.
39. Shakoury-Elizeh M, Tiedeman J, Rashford J, Ferea T, Demeter J, Garcia E, Rolfes R, Brown PO, Botstein D, Philpott CC. 2004. Transcriptional Remodeling in Response to Iron Deprivation in *Saccharomyces cerevisiae*. *MBoC* 15:1233–1243.
40. Lemire BD, Oyedotun KS. 2002. The *Saccharomyces cerevisiae* mitochondrial succinate:ubiquinone oxidoreductase. *Biochimica et Biophysica Acta (BBA) - Bioenergetics* 1553:102–116.
41. Chen XJ, Wang X, Kaufman BA, Butow RA. 2005. Aconitase Couples Metabolic Regulation to Mitochondrial DNA Maintenance. *Science* 307:714–717.
42. Kaplan CD, Kaplan J. 2009. Iron Acquisition and Transcriptional Regulation. *Chem Rev* 109:4536–4552.
43. Lill R, Hoffmann B, Molik S, Pierik AJ, Rietzschel N, Stehling O, Uzarska MA, Webert H, Wilbrecht C, Mühlenhoff U. 2012. The role of mitochondria in cellular iron–sulfur protein biogenesis and iron metabolism. *Biochimica et Biophysica Acta (BBA) - Molecular Cell Research* 1823:1491–1508.
44. Ramos-Alonso L, Romero AM, Soler MÀ, Perea-García A, Alepuz P, Puig S, Martínez-Pastor MT. 2018. Yeast Cth2 protein represses the translation of ARE-containing mRNAs in response to iron deficiency. *PLoS Genet* 14:e1007476.

45. Puig S, Askeland E, Thiele DJ. 2005. Coordinated remodeling of cellular metabolism during iron deficiency through targeted mRNA degradation. *Cell* 120:99–110.
46. Gerwien F, Skrahina V, Kasper L, Hube B, Brunke S. 2018. Metals in fungal virulence. *FEMS Microbiol Rev* 42:fux050.
47. Gomez M, Pérez-Gallardo RV, Sánchez LA, Díaz-Pérez AL, Cortés-Rojo C, Meza Carmen V, Saavedra-Molina A, Lara-Romero J, Jiménez-Sandoval S, Rodríguez F, Rodríguez-Zavala JS, Campos-García J. 2014. Malfunctioning of the Iron–Sulfur Cluster Assembly Machinery in *Saccharomyces cerevisiae* Produces Oxidative Stress via an Iron-Dependent Mechanism, Causing Dysfunction in Respiratory Complexes. *PLoS ONE* 9:e111585.
48. She X, Zhang P, Gao Y, Zhang L, Wang Q, Chen H, Calderone R, Liu W, Li D. 2018. A mitochondrial proteomics view of complex I deficiency in *Candida albicans*. *Mitochondrion* 38:48–57.
49. Li Y, Zhang Y, Zhang C, Wang H, Wei X, Chen P, Lu L. 2020. Mitochondrial dysfunctions trigger the calcium signaling-dependent fungal multidrug resistance. *Proc Natl Acad Sci USA* 117:1711–1721.
50. Simm C, Weerasinghe H, Thomas DR, Harrison PF, Newton HJ, Beilharz TH, Traven A. 2022. Disruption of Iron Homeostasis and Mitochondrial Metabolism Are Promising Targets to Inhibit *Candida auris*. *Microbiol Spectr* 10:e00100-22.
51. Lv Q, Yan L, Wang J, Feng J, Gao L, Qiu L, Chao W, Qin Y-L, Jiang Y. 2023. Combined Transcriptome and Metabolome Analysis Reveals That the Potent Antifungal Pyrylium Salt Inhibits Mitochondrial Complex I in *Candida albicans*. *Microbiol Spectr* 11:e03209-22.
52. Jordá T, Barba-Aliaga M, Rozès N, Alepuz P, Martínez-Pastor MT, Puig S. 2022. Transcriptional regulation of ergosterol biosynthesis genes in response to iron deficiency. *Environ Microbiol* 24:5248–5260.
53. Kitada K, Yamaguchi E, Arisawa M. 1996. Isolation of a *Candida glabrata* centromere and its use in construction of plasmid vectors. *Gene* 175:105–108.

54. Brachmann CB, Davies A, Cost GJ, Caputo E, Li J, Hieter P, Boeke JD. 1998. Designer deletion strains derived from *Saccharomyces cerevisiae* S288C: a useful set of strains and plasmids for PCR-mediated gene disruption and other applications. *Yeast* 14:115–132.
55. Giaever G, Chu AM, Ni L, Connelly C, Riles L, Véronneau S, Dow S, Lucau-Danila A, Anderson K, André B, Arkin AP, Astromoff A, El Bakkoury M, Bangham R, Benito R, Brachat S, Campanaro S, Curtiss M, Davis K, Deutschbauer A, Entian K-D, Flaherty P, Foury F, Garfinkel DJ, Gerstein M, Gotte D, Güldener U, Hegemann JH, Hempel S, Herman Z, Jaramillo DF, Kelly DE, Kelly SL, Kötter P, LaBonte D, Lamb DC, Lan N, Liang H, Liao H, Liu L, Luo C, Lussier M, Mao R, Menard P, Ooi SL, Revuelta JL, Roberts CJ, Rose M, Ross-Macdonald P, Scherens B, Schimmack G, Shafer B, Shoemaker DD, Sookhai-Mahadeo S, Storms RK, Strathern JN, Valle G, Voet M, Volckaert G, Wang C, Ward TR, Wilhelmy J, Winzeler EA, Yang Y, Yen G, Youngman E, Yu K, Bussey H, Boeke JD, Snyder M, Philippsen P, Davis RW, Johnston M. 2002. Functional profiling of the *Saccharomyces cerevisiae* genome. *Nature* 418:387–391.
56. Berg HS, Seterdal KE, Smetop T, Rozenvalds R, Brandtzaeg OK, Vehus T, Lundanes E, Wilson SR. 2017. Self-packed core shell nano liquid chromatography columns and silica-based monolithic trap columns for targeted proteomics. *Journal of Chromatography A* 1498:111–119.
57. Ramos-Alonso L, Holland P, Le Gras S, Zhao X, Jost B, Bjørås M, Barral Y, Enserink JM, Chymkowitch P. 2023. Mitotic chromosome condensation resets chromatin to safeguard transcriptional homeostasis during interphase. *Proc Natl Acad Sci USA* 120:e2210593120.
58. Martin M. 2011. Cutadapt removes adapter sequences from high-throughput sequencing reads. *EMBnet j* 17:10.
59. Dobin A, Davis CA, Schlesinger F, Drenkow J, Zaleski C, Jha S, Batut P, Chaisson M, Gingeras TR. 2013. STAR: ultrafast universal RNA-seq aligner. *Bioinformatics* 29:15–21.

60. Anders S, Pyl PT, Huber W. 2015. HTSeq—a Python framework to work with high-throughput sequencing data. *Bioinformatics* 31:166–169.
61. Anders S, Huber W. 2010. Differential expression analysis for sequence count data. *Genome Biol* 11:R106.
62. Love MI, Huber W, Anders S. 2014. Moderated estimation of fold change and dispersion for RNA-seq data with DESeq2. *Genome Biol* 15:550.
63. Benjamini Y, Hochberg Y. 1995. Controlling the False Discovery Rate: A Practical and Powerful Approach to Multiple Testing. *Journal of the Royal Statistical Society: Series B (Methodological)* 57:289–300.
64. Garcia I, Orellana-Muñoz S, Ramos-Alonso L, Andersen AN, Zimmermann C, Eriksson J, Bøe SO, Kaferle P, Papamichos-Chronakis M, Chymkowitz P, Enserink JM. 2021. Kell is a phosphorylation-regulated noise suppressor of the pheromone signaling pathway. *Cell Rep* 37:110186.
65. James J, Fiji N, Roy D, Andrew Mg D, Shihabudeen MS, Chattopadhyay D, Thirumurugan K. 2015. A rapid method to assess reactive oxygen species in yeast using H₂ DCF-DA. *Anal Methods* 7:8572–8575.

Figure legends

Figure 1. Identification of the ColA analog NR-6226C as an antifungal agent with a potential therapeutic window. **A**, ColA compound structure. **B**, Heat map of relative proliferation of several WT and echinocandin- and azole-resistant *Candida* strains. The indicated strains were incubated in SD media with increasing concentrations of ColA for 24h, after which proliferation was analyzed by measuring the OD₆₀₀. Data were normalized to DMSO-treated control samples. **C**, Heat map of *Candida* proliferation in the presence of the ColA analogs NR-6226C, NR-6226K, and NR-6226V.

The indicated fungal strains were incubated in the presence of increasing compound concentrations in YNB-iron media for 24h, after which proliferation was measured using OD₆₀₀. **D**, Human HEK-293 and HS-5 cells are considerably less sensitive to ColA analogs than *Candida spp.* Cells were incubated with increasing concentrations of the indicated compounds for 24 hrs, after which relative cell viability was assayed using CellTiter-Glo. Data were normalized to DMSO controls.

Figure 2. NR-6226C is an iron chelator that inhibits *Candida* proliferation by sequestering iron. **A**, ColH, which does not bind iron, does not inhibit growth of *C. glabrata*. Fungal cells were treated with ColA, ColH and NR-6226C in SD media for 24h, after which proliferation was measured using OD₆₀₀. Data were normalized to DMSO control samples. **B**, NR-6226C selectively binds Fe²⁺ *in vitro*. The interaction of NR-6226C with increasing concentrations of the indicated metal ions was determined. *n*=3. **C**, Exogenous Fe²⁺ rescues the antiproliferative effect of NR-6226C. Wild-type *C. glabrata* was incubated for 24h with NR-6226C in presence of 5 μM CaCl₂, CuCl₂, FeCl₂, FeCl₃, MgCl₂, MnCl₂, or ZnCl₂, respectively. Proliferation was measured using OD₆₀₀ and values were normalized to DMSO controls. Resulting EC₅₀ values were calculated and statistical significance was analyzed using unpaired two-sample t-tests in R, ****P*≤0.001; ***P*≤0.01; *n*=3.

Figure 3. NR-6226C treatment induces an iron starvation response. **A**, Wild-type *C. glabrata* cells were incubated with either DMSO or 10 μM NR-6226C for 1h, after which RNA levels were analyzed by RNA-seq. The volcano plot shows transcripts below (blue) or above (red) a 1.4-fold change (log₂-fold change <-0.5 or >0.5) and an adjusted p-value <0.01 threshold in NR-6226C-treated cells relative to the DMSO control. **B**, GO analysis of genes that are either upregulated (top) or downregulated (bottom) by NR-6226C treatment. **C**, Validation of a panel of selected genes by RT-qPCR shows that NR-6226C treatment activates the transcription factor Aft1. Wild-type or *aft1Δ S. cerevisiae* strains were treated with DMSO, 10 μM NR-6226C, or 10 μM Dp44mT for 24 hrs, after which RNA levels were analyzed by RT-qPCR. cDNA concentrations were normalized to

DMSO controls, followed by calculation of the means, sample standard deviations and statistical significance using unpaired two-sample t-tests in base R. *** $P \leq 0.001$; ** $P \leq 0.01$; *ns*, not significant; $n=3$.

Figure 4. NR-6226C treatment impairs mitochondrial functions and induces ROS formation.

A, *C. glabrata* cells were preloaded with the fluorescent ROS sensor H₂DCF-DA, after which changes in fluorescence were measured after 3, 6, and 24 hrs after exposure to either DMSO, 30 μ M NR-6226C, or 75 μ M H₂O₂. **B**, *C. glabrata* cells were treated for 24 hrs with DMSO, 0.625 μ M, 2.5 μ M or 10 μ M NR-6226C, after which relative ATP levels were measured by luminescence using CellTiter-Glo. An equal number of cells was harvested for each experiment, and luminescence was normalized to DMSO. **C**, *C. glabrata* cells were treated with DMSO, 2.5 μ M or 10 μ M NR-6226C for 24 hrs, after which they were loaded with MitoTracker and imaged using a Zeiss LSM 710 microscope. *** $P \leq 0.001$; ** $P \leq 0.01$; *ns*, not significant; $n=3$.

Figure 5. NR-6226C treatment improves the survival of *C. glabrata*-infected *G. mellonella* larvae.

A,B, *G. mellonella* larvae were infected with either *C. glabrata* 2001HT (A) or *C. glabrata* FL-256 (B), and treated with either DMSO or 30 μ M NR-6226C, after which overall survival was monitored over time. Survival curves were generated using the Kaplan-Meier formula. Statistical significance was calculated using log-rank tests in R.

Figure 6. Fluconazole and NR-6226C have a synergistic effect.

A, Dose response matrix of *C. albicans* WT cells treated in combination with Fluconazole and NR-6226C. Relative cell numbers were quantified as described in Figure 1B. **B**, 3D visualization of the synergy matrix shown in (A) using the Bliss synergy model. Panel A and B were generated using SynergyFinder in R.

Supplemental Figure Legends

Supplemental Figure S1. Collismycin-related compound names and structures.

Supplemental Figure S2. Compound screening. Heatmap of relative *Candida* proliferation after treatment for 24 hrs with the indicated concentrations of Collismycin analogs. Cell growth was measured using OD₆₀₀. Data were normalized to cells treated with DMSO.

Supplemental Figure S3. A, B, Proliferation curves and EC₅₀ values of *Candida spp* treated for 24 hrs with either Collismycin A (A) or with NR-6226C (26C), NR-6226K (26K), NR-6226V (26V) (B). Cell growth was measured using OD₆₀₀. **C,** Proliferation curves and EC₅₀ values of HEK-293 and HS-5 cells after treatment for 24 hrs with either NR-6226C, NR-6226K, or NR-6226V. Closed circles indicate technical replicates, lines indicate biological replicates. EC₅₀ values were calculated as described in Figure 2C.

Supplemental Figure S4. HEK-293 and HS-5 cells were treated with either DMSO, 100 μM NR-6226C, or 100 μM NR-6226K and imaged using the Incucyte Live-Cell Analysis System. Red arrows indicate compound precipitation.

Supplemental Figure S5. Mass spectroscopy of NR-6226C on its own and or bound to Fe²⁺.

Supplemental Figure S6. A, Volcano plot showing commonly regulated genes based on the threshold for the calculated mean difference and standard deviation in log fold-change between *C. glabrata* cells treated with either NR-6226C or Dp44mT (log₂-fold change <-0.683 or >0.635).

B,C, Venn diagram showing the count and respective percentages of up-regulated (B) or down-regulated (C) genes between *C. glabrata* cells treated with either NR-6226C or Dp44mT.

Supplemental Figure S7. A, Growth curves of *C. glabrata* cells pre-loaded with the H₂DCFDA and then treated with DMSO, 30 μ M NR-6226C, or 75 μ M H₂O₂. Cell proliferation was measured and corrected using OD₆₀₀. **B**, EC₅₀ values of *C. glabrata* cells treated with NR-6226C alone within a concentration range of 0.78 μ M-100 μ M, or with NR-6226C in combination with 1 μ M, 5 μ M or 10 μ M N-Acetyl-L-Cysteine (NAC). Samples were normalized to DMSO controls, followed by EC₅₀ estimation. Statistical significance was calculated using unpaired two-sample t-tests. *** $P \leq 0.001$; ** $P \leq 0.01$; *ns*, not significant; $n=3$.

Supplemental Figure S8. A, C, E, G, Bliss synergy scores of *Candida spp.* treatment with Fluconazole and NR-6226C combination. **B, D, F**, Dose response matrix of *Candida glabrata* cells treated in combination with Fluconazole and NR-6226C. Relative cell numbers were quantified as described in Figure 1B. Dose response matrices and synergy scores were obtained using SynergyFinder in R.

We thank the reviewers for taking their time to provide important feedback on our manuscript, which helped us improve the manuscript substantially. As described below, we have incorporated nearly all suggestions and we have added several new experiments.

Reviewer comments:

Reviewer #1 (Comments for the Author):

In this manuscript, the authors identify NR-6226C, a structural analog of collismycin A, as capable of inhibiting the growth of *Candida albicans* and *Candida glabrata*. The authors go on to characterize its effect on *C. glabrata* physiology through RNA-seq, making a case that iron chelation is the mechanism underlying its antifungal activity. Additionally, the authors show a reduction in mortality in a *Galleria* model of *C. glabrata* virulence, and that NR-6226C synergizes with fluconazole using a Bliss model for synergy.

Comments and suggestions:

Lines 110-113: You write "the clinical isolate FL-256 112 showing the greatest sensitivity." The heat maps in Figure 1b do not appear to support this (WT *C. glabrata* appears to be the most sensitive). Going into figure S3 to see the raw data used to calculate the EC50 and generate the heat map, there appears to be wide variability between biological replicates for FL-256. Were statistical methodologies employed that accounted for the variability in these data when calculating the EC50 for FL-256 112?

We agree with the reviewer and removed that specific statement.

Line 133-135: You state "NR-6226C and NR-6226K have antifungal activity exceeding that of CoIA, with relatively low toxicity against mammalian cells." Can that be said of NR-6226K, given that you have reason to believe it was insoluble in your toxicity assays, and is likely not bioavailable?

The reviewer is correct and we have removed the statement about low toxicity against mammalian cells.

The sentence now reads: "Together, these results show that the CoIA analogs NR-6226C and NR-6226K have antifungal activity exceeding that of CoIA."

Figure S2 is probably extraneous. Much of what it contains is negative data, and the data it contains from the hits from the compound screen are duplicated elsewhere in the manuscript. Similarly, it seems that Figure 1B and 1C represent data from the same set of experiments; if so, they could be condensed into one to let the reader know comparisons across them can be made.

While we agree with the reviewer in principle, we prefer to maintain the original figure organization for esthetic reasons (we found that combining all data makes it difficult to navigate the figure and the text becomes somewhat obfuscated). More importantly, even though Fig. S2 mostly contains negative data, there are differences between compounds that could be of interest to other researchers.

The data in Figure S6 are important to making the case that NR-6226C induces an iron starvation response. I would consider moving it from the supplement into the body of the manuscript.

Good point. The data have been moved from Fig. S6 to Figure 3.

Conventionally, effective antimicrobial concentrations and synergy are discussed in the literature in terms of MIC and FIC. EC50, in this case, is probably equivalent to MIC50, but the authors use a Bliss model for synergy in this manuscript. Is this an appropriate synergy test for antimicrobials, where the effective concentration is determined by inhibition of the growth of a whole organism, rather than inhibition of a molecular target?

*As the reviewer probably knows, identifying synergy is not quite as easy as it seems and there is a heated debate in the literature regarding which model performs best. The reason for using the Bliss model (which requires EC50 values) is that there are several problems with Loewe-based synergy models (such as FIC). For instance, the Loewe model is based on the assumption that a drug can only be additive when combined with itself, and therefore this model of response additivity is only correct if drugs have linear dose-effect curves with zero intercepts. However, most dose-effect curves in pharmacology and physiology are curvilinear (1), which was -somewhat ironically- already recognized by Loewe himself (2). Furthermore, the Loewe model frequently suffers from curve fitting problems, where experimental variation in the dose-response curves cannot always be fitted with logistic functions (3). Although not perfect, the Bliss model circumvents these issues. The Bliss model has been used to identify synergy between drugs in whole organisms, including *Plasmodium falciparum* (4). Therefore, we decided to use the Bliss model in SynergyFinder, which originally was developed for identification of synergistic drug interactions in hematopoietic cells (which, to a certain extent, behave like single-cell whole organisms).*

Minor comments and suggestions:

Lines 36 and 41: Azoles and echinocandins are antifungals, not antibiotics. Consider "antimicrobials," as antibiotics conventionally refers to antibacterial compounds.

Done

Line 50: the data presented do not differentiate between fungicidal activity and fungistatic activity. "Killing" is probably inappropriate here. Consider "inhibition of."

Done

Sentence beginning on line 212: Suggest changing "Candida sp." to *Candida glabrata*.

Done

Consider clarifying in the subheading on line 259 that synergy between NR 6226C and fluconazole exists for *C. albicans*, and not *C. glabrata*.

We changed the subheading into: "Synergistic antifungal effect of NR-6226C and fluconazole against C. albicans"

It would be great if Table S2 also included the fold changes of those altered transcripts, but that is a wishlist item.

That is indeed a good idea, because it makes the data more accessible for other researchers. The table has been updated accordingly.

Reviewer #2 (Comments for the Author):

The paper describes the antifungal effects of a lead compound that shows promise as a potential new antifungal. Collismycin A derivatives were screened and one compound, NR-6226C, inhibited the proliferation of WT *C. albicans* and *C. glabrata*, and two antifungal-resistant *C. glabrata* strains, while the proliferation of two human cell lines was much less impaired, suggesting there may be a therapeutic window for the compound. The authors hypothesize that NR-6226C exerts its antifungal effects through iron chelation, inducing an iron starvation response and inhibiting mitochondrial function, although the experimental evidence for this mechanism is less robust. Using a *G. mellonella* model, the authors demonstrate that NR-6226C has antifungal effects *in vivo*, significantly improving the survival of larvae infected with either WT or azole-resistant *C. glabrata*, although at a much higher concentration of NR-6226C than that required to inhibit fungal proliferation *in vitro*. Finally, NR-6226C and fluconazole have a synergistic effect against *C. albicans* *in vitro*, although this synergy is not seen in *C. glabrata*, which the authors hypothesize may be due to *C. glabrata*'s low susceptibility to fluconazole. Overall, the paper describes an intriguing potential new antifungal compound; the text could benefit from some minor edits for clarity. The degree of certainty regarding the compound's mechanism of action may be somewhat overstated and is arguably the paper's greatest shortcoming; however, the authors' findings appear to be experimentally sound overall, and the results still represent a useful contribution to the field.

The following points are suggestions to improve the clarity of the paper.

1. Antibiotic vs antimicrobial, antifungal: In several places throughout the paper (such as lines 36, 41, 85), the term "antibiotic" is used to refer to a compound with antifungal properties; as "antibiotic" is most commonly used to mean "antibacterial," the term "antimicrobial" might be more appropriate.

This was also mentioned by Reviewer #1 and has been altered.

2. Lines 116-118: I think the authors mean either an EC₅₀ 12x smaller (not greater) than, or an inhibitory effect 12x greater than ColA. It would be helpful to give the range of EC₅₀s in the text (1.00 +/- 0.49 to 4.78 +/- 0.53) to make it easier for the reader to compare these values to those given for human cell lines in the next paragraph.

We thank the reviewer for catching this and we made the correction. We decided not to add the numbers to the text, because adding these values for the different yeast strains made the text more cumbersome to read.

3. Line 156: Do the authors have data showing that NR-6226C binds Fe³⁺ in a 2:1 complex? I would suggest either including it in Supplemental Fig 5 or saying "data not shown." Or perhaps the authors are referring to the data shown in Fig 2B and simply mean that it binds to Fe³⁺? It's a very minor point which the authors may choose to ignore, I was simply confused about whether or not they meant to indicate the binding ratio was the same.

We presently do not have data demonstrating that NR-6226C binds Fe³⁺ in a 2:1 complex. However, we realized that the original description of the data could cause confusion, and therefore we rewrote this sentence as follows: "These nano liquid chromatography experiments showed that similar to ColA, NR-6226C forms a 2:1 compound-iron complex, where it preferentially binds Fe²⁺ in a concentration-dependent manner (Fig. 2B and Suppl. Fig. S5). NR-6226C can also bind Fe³⁺, but to a much lesser extent than Fe²⁺ (Fig. 2B)."

4. The authors could consider softening the language when stating their conclusions about NR-6226C being an iron chelator (for example, on lines 149 and 166, could change "showing" and "indicate" to "suggesting"/"suggest"). While their hypothesis is plausible, at least in 2D, the spacing of the three putative iron-binding N atoms in NR-6226C looks rather different from that in ColA (2 vs 3 carbons between N atoms); the lack of the third N is not the only structural difference between ColA and ColH; and the fact that CuCl₂ attenuates the

effects of 26C as effectively as FeCl₂ (and ZnCl₂ as effectively as FeCl₃) may indeed be due to mismetallation, but it is also possible that 26C acts via a mechanism other than iron binding.

We have modified the final sentence of the paragraph as follows: "Overall, these findings are consistent with the idea that NR-6226C inhibits the proliferation of Candida strains mainly through iron chelation, although we cannot exclude the possibility that it acts through alternative mechanisms in vivo."

5. Treatment with NR-6226C induces an iron starvation response: was there a particular reason for choosing the experimental conditions for the mRNA expression experiments? The authors explain their reason for choosing 1 h (as opposed to 24 h) for treating the *C. glabrata* cells, but not how they chose 10 μ M 26C (quite a bit higher than the EC₅₀). I assumed that 5 μ M Dp44mT was chosen because it was a concentration that was known to be effective. However, they then treated with *S. cerevisiae* for 24 h with 10 μ M 26C or 10 μ M Dp44mT. If there was a particular rationale for this choice, it might be helpful to clarify that in the text.

There was no particular reason for choosing 10 μ M except that we wanted to be certain that the compounds were present at saturating concentrations for the duration of the experiment, because we have not yet determined the half-life of the compound.

6. The authors might wish to say something about Fig 3B in the text. In line 191, I believe they are referring to Fig 3C.

Corrected.

7. Line 211: I believe this should say Fig 3D, not Fig 3C.

Corrected.

8. Treatment with NR-6226C induces ROS formation: it would be helpful for the authors to explain why they chose to use 30 μ M 26C, which is much higher than either the EC₅₀ or the concentration used in the gene expression experiments. It seems a bit odd that a transcriptional response to oxidative damage could be detected after 1h treatment with 10 μ M 26C, but that no significant change in ROS formation was detectable after 3 or 6 h treatment with 30 μ M 26C, unless the cells are mounting such an effective response that the ROS production is not detectable, or the cells have simply died. According to supplemental fig S7A (not referred to in the text), it is clear that there is little, if any, proliferation occurring in cells treated with 26C or H₂O₂; are the cells still viable? For the experiments shown in Figs 4B and 4C, it is stated that an equal number of cells was harvested under each condition, yet in Fig 4C there appear to be far fewer cells in the 10 μ M 26C condition than the 2.5 μ M 26C condition. If the cells treated with 30 μ M 26C mostly died, that could have contributed to the difficulty in detecting ROS formation in Fig 4A. Similarly, it is unclear whether the data in Fig 4B represent a loss of mitochondrial activity and ATP production specifically, or whether this simply demonstrates that the cells are no longer viable (for reasons that may or may have been caused by mitochondrial dysfunction). Regardless of whether or not the cells survived, to support a mechanistic argument, it would have been more convincing to see an effect on cellular function at a concentration and timepoint at or below the EC₅₀ and 24 hours. Negative data (no measurable effect under conditions that inhibit proliferation) are still important, but would have supported a different argument.

This is a very important point indeed, because studying metabolic processes in dead cells is not very interesting – at least not in this context. We performed several new experiments with concentration courses and timecourses, looking at mitochondrial activity, ATP levels and overall cell viability. These experiments show that mitochondrial activity and ATP levels are significantly reduced already after 6 hrs, but that cells are still viable even after 24hrs of NR-6226C treatment. We conclude that the observed loss of mitochondrial activity and ATP production are not simply due to the fact that we are studying dead cells. These data have been included in the manuscript and we added a paragraph to describe these results.

9. In vivo infection model: It is understandable that a higher concentration may be required in an animal model, but it might be useful to acknowledge that the authors chose to use a higher concentration of 26C in this model than was needed to inhibit proliferation in vitro.

*One of the reasons for choosing a high concentration of NR-6226C was to test whether it was toxic to the larvae (it was well tolerated). Nonetheless, we have modified the text as follows: "In brief, *G. mellonella* larvae were infected with WT *C. glabrata*, followed by incubation in the presence of either DMSO (untreated control) or 30 μ M NR-6226C, a concentration well above the in vitro IC₅₀ for fungal proliferation."*

10. Lines 311-312: The mechanism of downregulation in response to iron depletion may indeed be similar in *C. albicans*, but I'm not sure I would say that there is a high degree of conservation between *S. cerevisiae* and *C. albicans*.

Agreed. We have modified the sentence accordingly.

11. Figure 2B: I believe there is an "l" missing in the word "metal" in the y-axis label.

Corrected.

12. Figure 3B: what is the difference between the two lines labeled "Function: Methyltransferase activity"? They both say "n=8," but have slightly different ratios.

This was a typo and has been corrected.

13. Other minor suggestions for clarity: please see suggested changes in Word document. In particular, there are a number of small but important suggested clarifications to the methods section.

These have been implemented (and we thank the reviewer for including the Word document, it was very helpful).

References

1. Geary, N. (2013) Understanding synergy. *Am J Physiol Endocrinol Metab* **304**, E237-253
2. Loewe, S. (1953) The problem of synergism and antagonism of combined drugs. *Arzneimittelforschung* **3**, 285-290
3. Yadav, B., Wennerberg, K., Aittokallio, T., and Tang, J. (2015) Searching for Drug Synergy in Complex Dose-Response Landscapes Using an Interaction Potency Model. *Comput Struct Biotechnol J* **13**, 504-513
4. Mott, B. T., Eastman, R. T., Guha, R., Sherlach, K. S., Siriwardana, A., Shinn, P., McKnight, C., Michael, S., Lacerda-Queiroz, N., Patel, P. R., Khine, P., Sun, H., Kasbekar, M., Aghdam, N., Fontaine, S. D., Liu, D., Mierzwa, T., Mathews-Griner, L. A., Ferrer, M., Renslo, A. R., Inglese, J., Yuan, J., Roepe, P. D., Su, X. Z., and Thomas, C. J. (2015) High-throughput matrix screening identifies synergistic and antagonistic antimalarial drug combinations. *Sci Rep* **5**, 13891

October 13, 2023

Prof. Jorrit Enserink
Oslo Universitetssykehus
Ullernchausseen 70
Oslo
Norway

Re: Spectrum02594-23R1 (Characterization of a selective, iron-chelating antifungal compound that disrupts fungal metabolism and synergizes with fluconazole)

Dear Prof. Jorrit Enserink:

Link Not Available

Sincerely,

Renato Kovacs

Journals Department
Reviewer comments:

Reviewer #2 (Comments for the Author):

This manuscript describes the antifungal effects of a lead compound that shows promise as a potential new antifungal. The authors show that it inhibits the proliferation of *C. albicans* and *C. glabrata*, and that it improves survival in a *G. mellonella* model, and synergizes with fluconazole. They suggest that it acts via iron chelation.

The authors have adequately addressed most of the reviewer comments. However, their assertion that they have demonstrated a decrease in mitochondrial function in living cells hinges upon the viability of the cells, which they have not clearly demonstrated. Thus, the authors' conclusions are not fully supported by their data. They could either address this issue using a

validated viability assay; use the CellTiter-Glo dat

This concern arises from the following points (the first two explain why viability needs to be established; point 3 is about the authors' attempt to address that concern).

1. ATP concentration is tightly regulated, which is why CellTiter-Glo is commonly used as a viability assay (including in *Candida* spp).

The authors grew cells in the presence of NR-6226C, harvested roughly the same number of cells per condition using OD600, and measured luminescence, which is directly proportional to ATP availability and thus usually to the number of live cells. Decreased luminescence in this assay is most commonly due to a decrease in the number of live cells. Alternatively, it could be due to a decrease in ATP concentration per cell due to impaired ATP synthesis; a severe impairment in electron transport chain function is one of the few scenarios that might plausibly lead to this outcome. This raised the issue of cell viability.

(While it still would not have directly supported the authors' proposed mechanism of action, a simpler and more quantitative approach would have been to grow the cells in the presence of drug in a 96-well plate for the desired length of time and then add CellTiter-Glo to demonstrate their compound reduces fungal cell viability.)

2. MitoTracker Red CMXRos is neither a direct nor quantitative measure of mitochondrial activity. It accumulates in active mitochondria in a membrane potential-dependent manner, where it binds to thiol groups and thus remains sequestered, even if membrane potential is subsequently lost. A lack of staining indicates that mitochondrial membrane potential has been lost, but not whether this is due to direct inhibition of mitochondrial function or because the cells are no longer viable.

(In spite of attempting to harvest the same number of cells per condition, there are far fewer cells per field of view at later timepoints and higher concentrations of drug. The difference in staining intensity for 1 vs 3 h DMSO demonstrates the difficulty in using this assay to quantify differences, though I do not disagree with the authors' statement that there is less staining after 24 h treatment with NR-6226C.

A more direct and quantitative measure of metabolic activity would have been an assay such as MTT or XTT reduction, which measures NADH, primarily from mitochondria. It can be performed in a 96-well plate, thus avoiding the semiquantitative cell harvesting step. However, this is often used as a viability assay because cells that are not metabolically active are typically not considered to be viable.)

3. To address the question of whether the observed reduction in ATP and MitoTracker staining was due to a loss of viability, the authors grew cells in the presence of drug, did a semi-quantitative harvest, and stained the cells with unconjugated FITC. FITC is an amine-reactive fluorescein derivative, and thus labels a variety of protein/peptide-containing molecules. The authors state that FITC can only pass the membrane of dead cells, and their heat-killed cells do appear to exhibit slightly more staining than their drug-treated cells. However, a reference for the use of unconjugated FITC as a vital dye would have been helpful; I could not readily find examples of this method in *Candida* spp, but have seen papers that used FITC to label live *Candida albicans* and *Candida glabrata* cells. I do not find this convincing evidence that the cells are viable.

There are a variety of well-validated vital dyes that can be used in *Candida* spp and quantified using a hemocytometer or flow cytometry.

(The quantification of live/dead staining of individual cells would be complementary to a quantitative assay such as MTT/XTT, which provide information about the entire population of cells.)

A few minor editorial/typographical issues were introduced during the revision process:

4. Line 132, the sentence "Together, these results show that the CoIA analogs...": because it has been edited in response to a reviewer comment, the sentence now seems out of place; would recommend moving this sentence to the previous paragraph or omitting it.

5. Line 198, "Fig 3E-F" should be "Fig 3E-G"

6. Because of the addition of Fig 3E-G, the text now discusses Fig 3C, then 3E-G, then 3D. For the sake of clarity, would suggest changing the order of panels in the figure or the order of paragraphs in the text so that the text discusses the panels in order.

7. Line 226 there is a typo: the text says "treatment for 4h...(Fig. 4A)" while the figure legend and label (and methods section) say 3h.

8. Methods section (lines 516 and 526) should be updated to include the new 1, 3, and 6h timepoints.

Staff Comments:

Preparing Revision Guidelines

Please return the manuscript within 60 days; if you cannot complete the modification within this time period, please contact me. If you do not wish to modify the manuscript and prefer to submit it to another journal, please notify me of your decision immediately so that the manuscript may be formally withdrawn from consideration by Microbiology Spectrum.

We appreciate the Reviewer's continued efforts to improve the quality of our manuscript. As described below, we performed several new experiments, which support our original conclusions.

Reviewer #2 (Comments for the Author):

This manuscript describes the antifungal effects of a lead compound that shows promise as a potential new antifungal. The authors show that it inhibits the proliferation of *C. albicans* and *C. glabrata*, and that it improves survival in a *G. mellonella* model, and synergizes with fluconazole. They suggest that it acts via iron chelation.

The authors have adequately addressed most of the reviewer comments. However, their assertion that they have demonstrated a decrease in mitochondrial function in living cells hinges upon the viability of the cells, which they have not clearly demonstrated. Thus, the authors' conclusions are not fully supported by their data. They could either address this issue using a validated viability assay; use the CellTiter-Glo dat

This concern arises from the following points (the first two explain why viability needs to be established; point 3 is about the authors' attempt to address that concern).

1. ATP concentration is tightly regulated, which is why CellTiter-Glo is commonly used as a viability assay (including in *Candida* spp).

The authors grew cells in the presence of NR-6226C, harvested roughly the same number of cells per condition using OD600, and measured luminescence, which is directly proportional to ATP availability and thus usually to the number of live cells. Decreased luminescence in this assay is most commonly due to a decrease in the number of live cells. Alternatively, it could be due to a decrease in ATP concentration per cell due to impaired ATP synthesis; a severe impairment in electron transport chain function is one of the few scenarios that might plausibly lead to this outcome. This raised the issue of cell viability.

(While it still would not have directly supported the authors' proposed mechanism of action, a simpler and more quantitative approach would have been to grow the cells in the presence of drug in a 96-well plate for the desired length of time and then add CellTiter-Glo to demonstrate their compound reduces fungal cell viability.)

We are not exactly sure what the issue is that the Reviewer raises in this comment. After internal deliberation we concluded that the Reviewer's concern probably boils down to the possibility that NR-6226C may have killed large numbers of fungal cells, thereby causing a reduction in CellTiter-Glo values (rather than by depletion of cellular ATP levels in living cells). This is a valid point. To determine whether NR-6226C kills cells after 24 hrs, we performed a cell viability assay with propidium iodide (see also point #3), and found that NR-6226C did not kill the cells even after 24h treatment with 10 μ M of NR-6226C (Fig. 4E). We conclude that the strong reduction in CellTiter-Glo values is not due to loss of cell viability. The simplest explanation of our results is that NR-6226C reduces cellular ATP levels in living cells.

2. MitoTracker Red CMXRos is neither a direct nor quantitative measure of mitochondrial activity. It accumulates in active mitochondria in a membrane potential-dependent manner, where it binds to thiol groups and thus remains sequestered, even if membrane potential is subsequently lost. A lack of staining indicates that mitochondrial membrane potential has been lost, but not whether this is due to direct inhibition of mitochondrial function or because the cells are no longer viable.

(In spite of attempting to harvest the same number of cells per condition, there are far fewer cells per field of view at later timepoints and higher concentrations of drug. The difference in staining intensity for 1 vs 3 h DMSO demonstrates the difficulty in using this assay to quantify differences, though I do not disagree with the authors' statement that there is less staining after 24 h treatment with NR-6226C.

A more direct and quantitative measure of metabolic activity would have been an assay such as MTT or XTT reduction, which measures NADH, primarily from mitochondria. It can be performed in a 96-well plate, thus avoiding the semiquantitative cell harvesting step. However, this is often used as a viability assay because cells that are not metabolically active are typically not considered to be viable.)

-We performed the XTT assay (Fig. 4C). The data show that NR-6226C treatment significantly reduces NADH levels. The results are fully consistent with the other data we provide in our manuscript.

-We agree that mitotracker alone is not sufficient to make hard conclusions about mitochondrial health; however, given that mitotracker fails to stain mitochondria that have lost their membrane potential, we do believe that the lack of mitochondrial staining after NR-6226C treatment is indicative of serious mitochondrial problems. Without membrane potential mitochondria are unable to synthesize ATP, which is consistent not only with our findings that ATP levels are lower in NR-6226C-treated cultures but also

with the XTT assay mentioned above. Many researchers will agree with that opinion. Therefore, we decided to keep the mitotracker data in the manuscript, because we believe these results do add value to the study. We performed a new experiment in which we quantified the intensity of mitotracker staining of the cells using a plate reader. These data are now shown in Fig. 4D. The microscopic images have been relegated to the Supplement, which will allow the reader to evaluate qualitative mitotracker data.

-[Regarding the comment about cell numbers in microscopic images: These are single images of random microscopic fields intended to provide representative images of mitotracker-stained cells; as the Reviewer surely knows, such microscopy images cannot be used as a proxy for actual cell numbers in corresponding cultures].

3. To address the question of whether the observed reduction in ATP and MitoTracker staining was due to a loss of viability, the authors grew cells in the presence of drug, did a semi-quantitative harvest, and stained the cells with unconjugated FITC. FITC is an amine-reactive fluorescein derivative, and thus labels a variety of protein/peptide-containing molecules. The authors state that FITC can only pass the membrane of dead cells, and their heat-killed cells do appear to exhibit slightly more staining than their drug-treated cells. However, a reference for the use of unconjugated FITC as a vital dye would have been helpful; I could not readily find examples of this method in *Candida* spp, but have seen papers that used FITC to label live *Candida albicans* and *Candida glabrata* cells. I do not find this convincing evidence that the cells are viable.

There are a variety of well-validated vital dyes that can be used in *Candida* spp and quantified using a hemocytometer or flow cytometry.

(The quantification of live/dead staining of individual cells would be complementary to a quantitative assay such as MTT/XTT, which provide information about the entire population of cells.)

As described above, we performed both the XTT assay and propidium iodide stains (the FITC experiments has been removed). After performing these new experiments our original conclusions remain unchanged.

We have not used a hemocytometer for our experiments, because, although we agree it does not provide accurate absolute cell counts, as long as OD600 measurements are within linear range and the instrument is well calibrated, within-experiment values are in fact comparable (Beal et al 2020, Communications Biology 3: 512) and can be used to equalize relative cell numbers between cultures.

A few minor editorial/typographical issues were introduced during the revision process:

4. Line 132, the sentence "Together, these results show that the CoIA analogs...": because it has been edited in response to a reviewer comment, the sentence now seems out of place; would recommend moving this sentence to the previous paragraph or omitting it.

Done

5. Line 198, "Fig 3E-F" should be "Fig 3E-G"

-Panel referencing has been updated (after reordering the other panels as suggested in comment #6)

6. Because of the addition of Fig 3E-G, the text now discusses Fig 3C, then 3E-G, then 3D. For the sake of clarity, would suggest changing the order of panels in the figure or the order of paragraphs in the text so that the text discusses the panels in order.

Done

7. Line 226 there is a typo: the text says "treatment for 4h...(Fig. 4A)" while the figure legend and label (and methods section) say 3h.

Corrected

8. Methods section (lines 516 and 526) should be updated to include the new 1, 3, and 6h timepoints.

The text has been modified

Re: Spectrum02594-23R2 (Characterization of a selective, iron-chelating antifungal compound that disrupts fungal metabolism and synergizes with fluconazole)

Dear Prof. Jorrit Enserink:

Your manuscript has been accepted, and I am forwarding it to the ASM production staff for publication. Your paper will first be checked to make sure all elements meet the technical requirements. ASM staff will contact you if anything needs to be revised before copyediting and production can begin. Otherwise, you will be notified when your proofs are ready to be viewed.

Sincerely,
Renato Kovacs
Editor
Microbiology Spectrum